# A partially nested cortical hierarchy of neural states underlies event segmentation in the human brain

Linda Geerligs[1]*, Dora Gözükara[1], Djamari Oetringer[1], Karen L Campbell[2], Marcel van Gerven[1], Umut Güçlü[1]

[1]Donders Institute for Brain, Cognition and Behaviour, Radboud University Nijmegen, Nijmegen, Netherlands; [2]Department of Psychology, Brock University, St. Catharines, Canada

**Abstract** A fundamental aspect of human experience is that it is segmented into discrete events. This may be underpinned by transitions between distinct neural states. Using an innovative data-driven state segmentation method, we investigate how neural states are organized across the cortical hierarchy and where in the cortex neural state boundaries and perceived event boundaries overlap. Our results show that neural state boundaries are organized in a temporal cortical hierarchy, with short states in primary sensory regions, and long states in lateral and medial prefrontal cortex. State boundaries are shared within and between groups of brain regions that resemble well-known functional networks. Perceived event boundaries overlap with neural state boundaries across large parts of the cortical hierarchy, particularly when those state boundaries demarcate a strong transition or are shared between brain regions. Taken together, these findings suggest that a partially nested cortical hierarchy of neural states forms the basis of event segmentation.

## Editor's evaluation

This article addresses the question of how the brain segments naturalistic events and the relationship between perceived event boundaries and neural pattern shifts. By applying an innovative analysis to a large, publicly available dataset, they observe evidence of different timescales of neural state shifts that correspond with perceived event bounds. These results will be of interest to cognitive neuroscientists investigating the relationship between neural states and event segmentation.

*For correspondence:
Linda.Geerligs@donders.ru.nl

Competing interest: The authors declare that no competing interests exist.

## Introduction

Segmentation of information into meaningful units is a fundamental feature of our conscious experience in real-life contexts. Spatial information processing is characterized by segmenting spatial regions into objects (e.g., *DiCarlo and Cox, 2007*). In a similar way, temporal information processing is characterized by segmenting our ongoing experience into separate events (*Kurby and Zacks, 2008*; *Newtson et al., 1977*). Segmentation improves our understanding of ongoing perceptual input (*Zacks et al., 2001a*) and allows us to recall distinct events from our past (*Flores et al., 2017*; *Sargent et al., 2013*; *Zacks et al., 2006*). Recent work has shown that the end of an event triggers an evoked response in the hippocampus (*Baldassano et al., 2017*; *Ben-Yakov and Henson, 2018*), suggesting that events form the basis of long-term memory representations. Events that are identified in written, auditory, and audiovisual narratives (movies) are often very similar across individuals and can be segmented hierarchically on different timescales (*Newtson and Rindner, 1979*; *Zacks et al., 2001a*). According to event segmentation theory (EST), perceptual systems spontaneously segment

activity into meaningful events as a side effect of predicting future information (*Zacks et al., 2007*). That is, event boundaries are perceived when predictions become less accurate, which can be due to a change in motion or features of the situation such as characters, causes, goals, and spatial location (*Zacks et al., 2009*). However, event boundaries are observed even when a change is predictable, suggesting that other mechanisms play a role (*Pettijohn and Radvansky, 2016*). One proposal is that experiences are grouped into categories (or event types), which we have learned previously. When a new event type is detected, an event boundary occurs (*Shin and DuBrow, 2021*).

While much is known about temporal event segmentation at a behavioral level, less is known about its neural underpinnings. A number of studies have investigated which brain regions show evoked responses around event boundaries. Although the exact regions vary across studies, commonly identified regions include the precuneus and medial visual cortex, as well as area V5 and the intraparietal sulcus (*Kurby and Zacks, 2018*; *Speer et al., 2007*; *Zacks et al., 2001b*; *Zacks et al., 2010*). Increased brain responses at event boundaries in these regions likely reflect updating processes that occur when shifting to a new event model (*Ezzyat and Davachi, 2011*). Recently, a different approach has been introduced to investigate the neural underpinnings of event segmentation (*Baldassano et al., 2017*). These authors applied a data-driven method, based on hidden Markov models (HMMs) to functional magnetic resonance imaging (fMRI) data obtained during movie watching to identify timepoints where brain activity in a particular region transitioned from one temporarily stable activity pattern to a different pattern. We refer to these periods of relative stability as neural states to distinguish them from subjectively perceived events (*Geerligs et al., 2021*; *Zacks et al., 2007*).

These neural states occur on different timescales across the cortical hierarchy, with short-lived states in early sensory regions and long-lasting states in higher-level regions such as the precuneus and angular gyrus (*Baldassano et al., 2017*), in line with previous observations of a temporal hierarchy of information processing in the brain (*Hasson et al., 2008*; *Honey et al., 2012*; *Lerner et al., 2011*; *Stephens et al., 2013*). Interestingly, for a set of four brain regions, *Baldassano et al., 2017* showed that neural state boundaries overlapped across different regions and with subjectively experienced event boundaries. These results suggest that neural state segmentation could be the source of perceived event boundaries and that states may be organized in a nested cortical hierarchy, such that the boundaries of faster states in regions lower in the cortical hierarchy are nested within the boundaries of slower regions higher up in the hierarchy. In such a nested hierarchy, each brain region integrates information within discretized neural states that may align with sensory units in the external environment (e.g., phonemes, words, sentences) and provide its output to the brain regions higher in the cortical hierarchy (*Nelson et al., 2017*), until neural states at the highest level of the hierarchy align with subjectively experienced events. This way information traveling up the hierarchy is gradually integrated into complex and long-lasting multimodal representations (*Hasson et al., 2015*). Although this is an intriguing hypothesis, the evidence for it is limited, as it has only been investigated in one previous study using four predefined regions of interest (*Baldassano et al., 2017*). In addition, it remains unknown which brain regions show neural state boundaries that align with perceived event boundaries and the temporal hierarchy of state segmentation remains unexplored across large parts of the cortex.

This study had two main aims. First, to investigate whether event segmentation is indeed underpinned by neural state segmentation occurring in a nested cortical hierarchy. Second, to characterize the temporal hierarchy of neural state segmentation across the entire cortex. If the brain segments ongoing input in a nested hierarchical fashion, we would expect to find especially long-lasting neural states in the frontal cortex, which is often considered the top of the cortical hierarchy (*Fuster, 2001*). We would also expect to find overlap between neural state boundaries and event boundaries across all levels of the cortical hierarchy, although this overlap should be most consistent for areas at higher levels of the hierarchy where the timescales of neural state segmentation should closely match the experienced timescale of events. Finally, we would expect that state boundaries are most strongly shared between groups of brain regions involved in similar cognitive functions (i.e., networks) and to a lesser extent between more distinct sets of brain areas.

To test these hypotheses, we used a novel data-driven state segmentation method that was specifically designed to reliably detect boundaries between distinct neural states (*Geerligs et al., 2021*). By using a large movie fMRI dataset from the Cam-CAN project (*Shafto et al., 2014*) that shows reliable stimulus-driven activity (i.e., significant inter-subject correlations) over nearly all cortical brain regions

(*Campbell et al., 2015*; *Geerligs and Campbell, 2018*), we were able to study neural state segmentation across the entire cortex for the first time. In comparison to previous work, we investigate state segmentation in a more focused and extensive way by identifying the degree of change moving from one neural state to the next (i.e., boundary strength) and examining relationships between neural state boundaries across functional networks.

## Results

To identify neural state boundaries, we applied an improved version of the greedy state boundary search (GSBS; *Geerligs et al., 2021*) to a large fMRI dataset in which 265 participants (aged 18–50 years) viewed an 8 min Alfred Hitchcock movie (*Shafto et al., 2014*). After hyperaligning the data (*Guntupalli et al., 2016*) and hemodynamic deconvolution, GSBS was applied to multi-voxel brain activity time courses from overlapping spherical searchlights covering the entire cortex. GSBS identifies a set of neural state boundaries for each searchlight and for different numbers of states ($k$). GSBS then uses the t-distance metric to identify the optimal number of state boundaries in each searchlight (*Geerligs et al., 2021*). This metric identifies the optimal number of state boundaries such that the Pearson correlations (across voxels) of timepoints within a state are maximized and correlations of timepoints in consecutive states are minimized. To optimize the validity and reliability of the neural states detected by GSBS, we improved the algorithm in several ways, as shown in the 'Supplementary methods' section in Appendix 1. Searchlights in which we were unable to identify reliable neural state boundaries were excluded from further analysis (see *Appendix 1—figure 5*). Searchlight-level results were projected to the voxel level by averaging results across overlapping searchlights.

The median duration of neural states differed greatly between brain regions, ranging from 4.5 s in the voxels with shortest states up to 27.2 s in the voxels with the longest states (see *Figure 1A*). Most voxels showed median state durations between 5.1 and 18.5 s per state. To determine whether regional differences in state duration were reliable, neural state boundaries were identified in two independent groups of participants. At the voxel level, there was a very high Pearson correlation between the median state durations of the two groups ($r = 0.85$; see *Figure 1A*). This correlation was lower when we computed it at the level of searchlights (i.e., before projecting to the voxel level; $r = 0.62$). This suggests that regional differences in neural state timescales are highly reliable across participant groups and that the variability present in specific searchlights can be reduced substantially by averaging across overlapping searchlights. The timing of neural state boundaries was not associated with head motion (see 'Supplementary results' in Appendix 1).

*Figure 1A* shows that there were particularly short neural states in visual cortex, early auditory cortex, and somatosensory cortex. State transitions were less frequent in areas further up cortical hierarchy, such as the angular gyrus, areas in posterior middle/inferior temporal cortex, precuneus, the anterior temporal pole, and anterior insula. Particularly long-lasting states were observed in high-level regions such as the medial prefrontal gyrus and anterior portions of the lateral prefrontal cortex, particularly in the left hemisphere.

We also investigated how the variability of state durations differed across the cortex. Because variability of state duration, as measured by the interquartile range (IQR), tends to increase as the median state duration increases, we used a nonparametric alternative to the coefficient of variation (IQR divided by the median). We found very pronounced variability in state durations in the middle and superior temporal gyri and the anterior insula, while the variability was consistently lower in all other cortical areas. This effect was highly reliable across the two independent groups of participants ($r = 0.84$ at voxel level; $r = 0.54$ at the searchlight level).

### Neural states and perceived event boundaries

In a nested cortical hierarchy, some boundaries at lower levels of the cortical hierarchy are thought to propagate to higher levels of the hierarchy until they are consciously experienced as event boundaries. Therefore, we would expect state boundaries to align with perceived event boundaries across all of the different levels of the hierarchy. The most consistent alignment would be expected in higher-level cortical areas where the number of states should more closely align with the number of perceived events. Event boundaries were determined by asking participants to indicate when they felt one event (meaningful unit) ended and another began (*Ben-Yakov and Henson, 2018*). To determine the



Figure 1. The cortical hierarchy of neural state durations. (**A**) The optimal number of states varied greatly across regions, with many shorter states in the primary visual, auditory, and sensorimotor cortices and few longer states in the association cortex, such as the medial and lateral prefrontal gyrus. These results are highly consistent across two independent groups of participants. Parts of the correlation matrices for two selected searchlights are shown in the insets for each of the groups, representing approximately 1.6 min of the movie. The white lines in these insets are the neural state boundaries that are detected by greedy state boundary search (GSBS). (**B**) The variability in state durations, as quantified by the interquartile range (IQR)/median state duration, was particularly high in the middle and superior temporal gyri and the anterior insula.

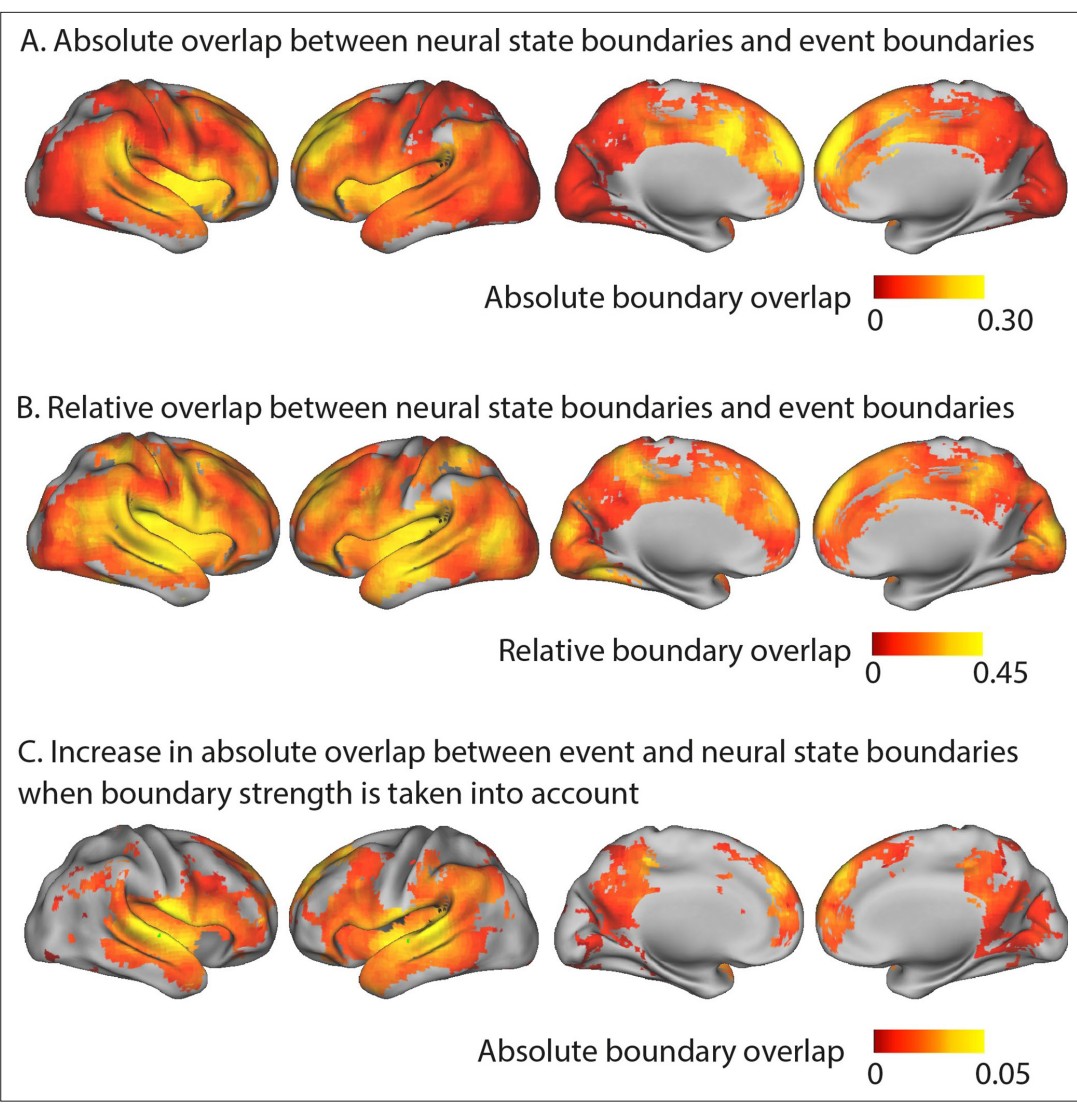

**Figure 2.** Overlap between neural state and event boundaries. (**A**) Absolute boundary overlap between neural state boundaries and the perceived event boundaries identified by a different group of participants outside the scanner. The metric is scaled between zero (expected overlap) and one (all neural state boundaries overlap with an event boundary). The medial prefrontal cortex, anterior insula, anterior cingulate cortex, and left superior and middle frontal gyrus show the strongest alignment between neural state boundaries and perceived event boundaries. (**B**) Relative overlap between neural state boundaries and perceived event boundaries (scaled w.r.t. the maximal possible overlap given the number of neural state boundaries). Regions in different parts of the cortical hierarchy (early and late) show a significant association between the neural state boundaries and perceived event boundaries. (**C**) Increase in absolute overlap between neural state boundaries and event boundaries when neural state boundaries are weighted by their strengths, in comparison to using binary boundaries (as in **A**). Regions with strong neural state boundaries (i.e., a large change between successive states) were more likely to overlap with perceived event boundaries than weak boundaries. Statistical analyses for (**A–C**) were based on data from 15 independent groups of participants, the depicted difference/overlap values were based on data in which all 265 participants were averaged together. All of the colored regions showed a significant association after false discovery rate (FDR) correction for multiple comparisons.

similarity between neural state boundaries and perceived event boundaries, we computed the absolute boundary overlap. This is defined as the number of timepoints where neural state and perceived event boundaries overlapped, scaled such that the measure is one when all neural state boundaries align with a perceived event boundary and zero when the overlap is equal to the overlap that would be expected given the number of boundaries.

We found that a large number of brain regions, throughout the cortical hierarchy, showed significant absolute boundary overlap between neural states and perceived event boundaries after false discovery rate (FDR) correction for multiple comparisons (see *Figure 2A*). In particular, we observed that the anterior cingulate cortex, dorsal medial prefrontal cortex, left superior and middle frontal gyrus, and anterior insula show the strongest absolute overlap between neural state boundaries and perceived boundaries. This suggests that neural state boundaries in these regions are most likely to underlie the experience of an event boundary.

The absolute boundary overlap is partly driven by regional differences in the number of neural state boundaries. However, our hypothesis of a nested cortical hierarchy suggests that regions in early stages of the cortical hierarchy, with many neural state boundaries, should also show overlap between neural state and perceived event boundaries. To correct for regional differences in the possibility for overlap, due to the differing number of neural state boundaries in a region, we computed the relative boundary overlap. The relative boundary overlap is scaled by the maximum possible overlap given the number of state and event boundaries. It is one when all perceived event boundaries coincide with a neural state boundary (even if there are many more neural states than events) or when all neural state boundaries coincide with an event boundary. A value of zero indicates that the overlap is equal to the expected overlap. This metric gives a different pattern of results, showing that regions across different levels of the cortical hierarchy (early and late) have strong overlap between neural states and perceived event boundaries (see *Figure 2B*). Regions early in the cortical hierarchy, such as the medial visual cortex, the medial and superior temporal gyri, and the postcentral gyrus, show strong relative overlap. The same is true for regions later in the hierarchy, including the anterior insula, most areas of the default mode network (DMN), including the medial frontal gyrus, anterior parts of the precuneus and the angular gyrus, and large parts of the lateral frontal cortex. These results suggest that there is overlap between event and neural state boundaries throughout the cortical hierarchy.

To understand why some neural state boundaries are associated with event boundaries and some are not, we investigated the degree of neural state change at each boundary. Specifically, we define boundary strength as the Pearson correlation distance between the neural activity patterns of consecutive neural states. We weighted each neural state boundary by the boundary strength and then recomputed the absolute overlap between neural state and event boundaries. Like before, the absolute overlap is one when all neural state boundaries align with a perceived event boundary and zero when the overlap is equal to the overlap that would be expected given the strengths of all neural state boundaries. However, after weighting boundaries by strength, given the same number of boundaries that overlap, the absolute overlap will be higher when the weaker neural state boundaries do not overlap with events and strong boundaries do overlap with event boundaries.

When we took the strength of neural state boundaries into account in this way, the absolute overlap with event boundaries increased compared to when we used a binary definition of neural state boundaries. This was observed particularly in the middle and superior temporal gyri, extending into the inferior frontal gyrus (see *Figure 2C*), but also in the precuneus and medial prefrontal cortex. This means that boundaries that coincided with a larger shift in brain activity patterns were more often associated with an experienced event boundary.

## Neural state networks

If neural state boundaries are organized in a nested cortical hierarchy, different brain regions would be expected to show substantial overlap in their neural state boundaries. Therefore, we investigated for each pair of searchlights whether the relative neural state boundary overlap was larger than would be expected by chance, given the number of boundaries in these regions. We observed that the overlap was significantly larger than chance for 85% of all pairs of searchlights, suggesting that neural state boundaries are indeed shared across large parts of the cortical hierarchy (see *Figure 3A*). Although the overlap was highly significant for the majority of searchlight pairs, we did not observe a perfectly nested architecture. If that were the case, the relative boundary overlap between searchlights would have been one. To make sure the observed relative boundary overlap between searchlights was not caused by noise shared across brain regions, we also computed the relative boundary overlap across two independent groups of participants (similar to the rationale of inter-subject functional connectivity analyses; *Simony et al., 2016*). We observed that the relative boundary overlap computed in this way was similar to the relative overlap computed within a participant group ($r = 0.69$; see

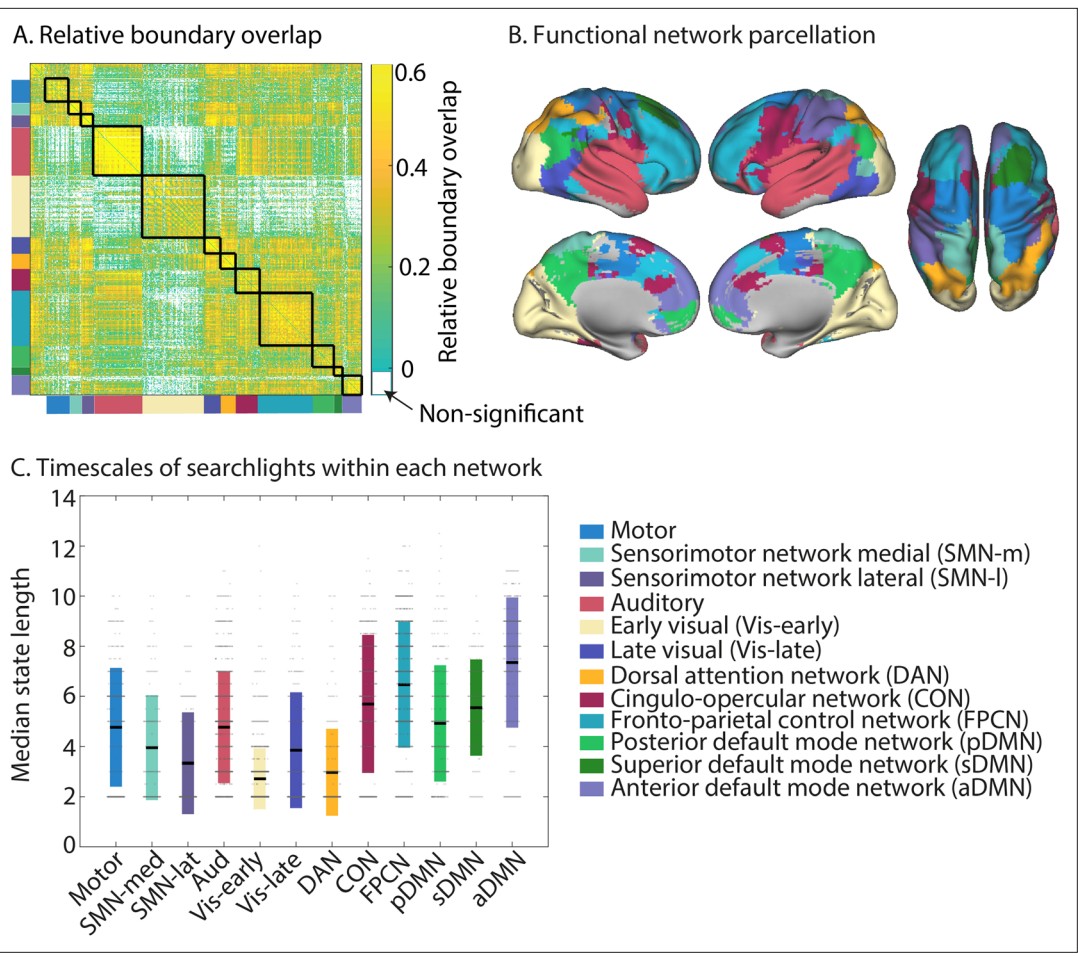

**Figure 3.** Neural state boundaries are shared within and across distinct functional networks that span the cortical hierarchy. (**A**) The relative boundary overlap between each pair of searchlights, ordered according the functional networks they are in. The black lines show the boundaries between functional networks. Searchlight pairs shown in white did not have significant relative boundary overlap after false discovery rate (FDR) correction for multiple comparisons. (**B**) Visualization of the detected functional networks. The network label at each voxel is determined by the functional network that occurs most often in all the searchlights that overlap with that voxel. (**C**) Visualization of the neural state durations within each network. Each searchlight is shown as a dot. The colored bars show the mean and 1 standard deviation around the mean for each network. The data shown in (**A–C**) are based on data averaged across all participants. The test for statistical significance in (**A**) was performed with data of 15 independent groups of participants. All of the colored regions in (**A**) showed a significant association after FDR correction for multiple comparisons.

Appendix 1—figure 6), suggesting that shared noise is not the cause of the observed regional overlap in neural state boundaries.

To investigate the overlap between regions in more detail, we identified networks of brain regions that shared state boundaries by computing the relative boundary overlap between each pair of searchlights and using consensus partitioning based on Louvain modularity maximization to identify networks (*Blondel et al., 2008*; *Lancichinetti and Fortunato, 2012*). We found that state boundaries were shared within long-range networks. Some of these networks resembled canonical networks typically identified based on (resting state) time-series correlations (see *Figure 3B*). To quantify this, we computed the proportion of searchlights overlapping with each of the networks defined by *Power et al., 2011* (see *Appendix 1—table 1*). We identified an auditory network that extended into regions involved in language processing in the left inferior frontal gyrus, a fronto-parietal control network (FPCN), a cingulo-opercular network (CON), and a motor network. The DMN we identified was fractionated into anterior, superior, and posterior components. It should be noted that all three of the DMN subnetworks include some anterior, superior, and posterior subregions; the names of these



**Figure 4.** Separate visualizations for each of the identified functional networks. The colors indicate the median state duration for each of the searchlights within the functional network. SMN, sensorimotor network; DMN, default mode network; FPCN, fronto-parietal control network; CON, cingulo-opercular network; DAN, dorsal attention network. The median state duration estimates are based on data averaged across all participants.

subnetworks indicate which aspects of the networks are most strongly represented. *Appendix 1— table 2* shows the overlap of each of these subnetworks with the anterior and posterior DMN as identified in *Campbell et al., 2013*. The sensorimotor network (SMN) we identified was split into a lateral and medial component. We also identified a network overlapping with the dorsal attention network (DAN), although the network we identified only covered posterior parts of the DAN but not the frontal eye fields. While the visual network is typically identified as a single network in functional connectivity studies, we observed two networks, roughly corresponding to different levels of the visual hierarchy (early and late). *Figure 3* visualizes for each voxel which functional network label occurs most frequently for the searchlights overlapping that voxel. In contrast, the full extent of each of the functional networks can be seen in *Figure 4*.

*Figure 3C* shows the average timescale within each of these functional networks. The networks with the longest state durations were the anterior DMN and the FPCN, while the early visual network,

lateral SMN, and DAN had particularly short state durations. Although regions within functional networks tended to operate on a similar temporal scale, we also observed a lot of variability in state duration within networks, particularly in the auditory network (see *Figure 3C*). Many networks also showed a clear within-network gradient of timescales, such as the auditory network, the SMNs, the posterior DMN, and the FPCN (see *Figure 4*). These results suggest that the relative boundary overlap between regions is not simply driven by a similarity in the number of states, but rather by a similarity in the state boundary timings. This is also supported by the results in *Appendix 1—figure 6*, showing that the relative boundary overlap was highly distinct from the absolute pairwise differences in median state duration ($r = -0.05$).

Although the networks we identified show overlap with functional networks previously identified in resting state, they clearly diverged for some networks (e.g., the visual network). Some divergence is expected because neural state boundaries are driven by shifts in voxel-activity patterns over time, rather than by the changes in mean activity that we typically use to infer functional connectivity. This divergence was supported by the overall limited similarity with the previously identified networks by *Power et al., 2011* (adjusted mutual information [aMI] = 0.39), as well as the differences between the correlation matrix that was computed based on the mean activity time courses in each searchlight and the relative boundary overlap between each pair of searchlights (*Appendix 1—figure 6*; $r = 0.31$). Interestingly, regions with strongly negatively correlated mean activity time courses typically showed overlap that was similar to or larger than the overlap expected by chance. Indeed, the relative boundary overlap between each pair of searchlights was more similar to the absolute Pearson correlation coefficient between searchlights ($r = 0.39$) than when the sign of the correlation coefficient was preserved ($r = 0.31$). This suggests that pairs of regions that show negatively correlated BOLD activity still tend to show neural state boundaries at the same time.

It should be noted that although the overlap in neural state boundaries was strongest for the searchlights that were part of the same functional networks, we also found a lot of evidence for the hypothesis that boundaries are shared across different levels of the cortical hierarchy (see *Figure 3A*). Overlap was particularly strong between all higher-order networks (DAN, CON, FPCN, and DMNs), as well as between the motor network and SMN. The sensorimotor, motor, and auditory networks also showed highly significant overlap with the higher-order networks. Lower levels of overlap were observed between the early visual network and all other networks (except the DAN), as well as between the auditory and the sensorimotor networks.

## Shared neural state boundaries and event boundaries

Previous research on event segmentation has shown that the perception of an event boundary is more likely when multiple features of a stimulus change at the same time (*Clewett et al., 2019*). When multiple sensory features changes at the same time, this could be reflected in many regions within the same functional network showing a state boundary at the same time (e.g., in the visual network when many aspects of the visual environment change), or in neural state boundaries that are shared across functional networks (e.g., across the auditory and visual networks when a visual and auditory change coincide). Similarly, boundaries shared between many brain regions within or across higher-level cortical networks might reflect a more pronounced change in conceptual features of the narrative (e.g., the goals or emotional state of the character). Therefore, we expect that in a nested cortical hierarchy neural state boundaries that are shared between many brain regions within functional networks, and particularly those shared widely across functional networks, would be more likely to be associated with the perception of an event boundary. To investigate this, we first weighted each neural state boundary in each searchlight by the proportion of searchlights within the same network that also showed a boundary at the same time. This is very similar to how we investigated the role of boundary strength above.

We found that when we took the within-network co-occurrence of neural state boundaries into account in this way, the absolute overlap with event boundaries increased compared to when we used a binary definition of neural state boundaries. So when a particular neural state boundary is shared with more regions within the same network, it is more likely to coincide with an event boundary (see *Figure 5A*). This effect was observed across all networks, except the early visual network. It was strongest for regions in the anterior DMN, the FPCN, and the auditory network and slightly less pronounced for regions in the posterior and superior DMN, as well as the CON. On a regional level,



**Figure 5.** Increase in absolute overlap between neural state boundaries and event boundaries when boundary co-occurence is taken into account. Increase in absolute overlap between neural state boundaries and event boundaries when neural state boundaries are weighted by the percentage of searchlights in the same functional networks or across the whole brain that also have a boundary at the same timepoint. (**A**) Within-network-weighted absolute overlap is compared to using binary boundaries (as in *Figure 2A*). (**B**) Whole-brain-weighted absolute overlap is compared to within-network-weighted absolute overlap. (**C**) The average absolute boundary overlap between events and neural states within each functional network. This is done for both binary boundaries, boundaries weighted by within-network co-occurrence, and boundaries weighted by whole-brain co-occurrence. A red star indicates a significant difference between binary boundaries and boundaries weighted by within-network co-occurrence. A blue star indicates a significant difference between boundaries weighted by whole-brain co-occurrence and boundaries weighted by within-network co-occurrence. The data shown in (**A–C**) are based on data averaged across all participants. The tests for statistical significance were performed with data of 15 independent groups of participants. All of the colored regions in (**A, B**) showed a significant association after false discovery rate (FDR) correction for multiple comparisons.

the strongest effects were observed within the precuneus, angular gyrus, medial prefrontal cortex, temporal pole, insula, superior temporal gyrus, and the middle frontal gyrus.

Next, we weighted each neural state boundary by the proportion of searchlights across the whole brain that showed a boundary at the same time. We investigated where absolute overlap for this whole-brain co-occurrence was stronger than when the neural state boundaries were weighted by



Figure 6. Increase in absolute overlap with event boundaries for shared vs. non-shared neural state boundaries. (**A**) For each pair of searchlights, we compare the relative boundary overlap between neural state boundaries between boundaries that are shared and boundaries that are not shared. In particular, for each pair of searchlights, the searchlight with the highest relative boundary overlap with events is used as the reference in the comparison. The white lines show the boundaries between functional networks. (**B**) shows the percentage of connections in (**A**) that show a significant increase in relative boundary overlap with events for shared vs. non-shared neural state boundaries. Here, the data are summarized to a network-by-network matrix for ease of interpretation. The data shown in this figure are based on data averaged across all participants. The tests for statistical significance were performed with data of 15 independent groups of participants. All of the colored regions in (**A**) showed a significant association after false discovery rate (FDR) correction for multiple comparisons.

within-network co-occurrence (see *Figure 5B*). This was the case specifically for the early and late visual networks, the DAN, the lateral and medial SMN networks, and the motor network. For the auditory network, the opposite effect was observed; whole-brain co-occurrence showed lower absolute overlap with event boundaries than within-network co-occurrence. On a regional level, increases in overlap with event boundaries were most pronounced in the medial parts of the occipital lobe, the supplementary motor area and precentral gyri, as well as the superior parietal gyri.

## A. Neural state boundaries and event boundaries in communities of timepoints



## B. Ratio of average (neural state) boundary occurence within each community versus average occurence across all timepoints within each network

**Figure 7.** Communities of timepoints identified with weighted stochastic block model (WSBM). This algorithm groups together timepoints that show similar boundary profiles (presence or absence of boundaries across searchlights). (**A**) Neural state boundaries are shown for each community per timepoint for each searchlight, grouped in functional networks. Boundaries shown in red coincide with an event boundary. (**B**) Per functional network, we show the ratio of the average state boundary occurrence within each community versus the average occurrence across all timepoints. The same is shown for the event boundaries. The data shown in this figure are based on data averaged across all participants.

To investigate the role of boundary co-occurrence across networks in more detail, we investigated for each pair of searchlights whether boundaries that are shared have a stronger association with perceived event boundaries as compared to boundaries that are unique to one of the two searchlights. We found that boundary sharing had a positive impact on overlap with perceived boundaries, particularly for pairs of searchlights within the auditory network and between the auditory network and the anterior DMN (see *Figure 6A and B*). In addition, we saw that neural state boundaries that were shared between the auditory network and the early and late visual networks, and the superior and posterior DMN were more likely to be associated with a perceived event boundary than non-shared boundaries. The same was true for boundaries shared between the anterior DMN and the lateral and medial SMN network and the posterior DMN. Boundary sharing between the other higher-level networks (pDMN, sDMN, FPCN, and CON) as well as between these higher-level networks and the SMN networks was also beneficial for overlap with event boundaries. On a regional level, the

strongest effects of boundary sharing were observed in the medial prefrontal cortex, medial occipital cortex, precuneus, middle and superior temporal gyrus, and insula (see *Figure 6B*). Analyses shown in the 'Supplementary results' section in Appendix 1 demonstrate that these increases in overlap for shared vs. non-shared boundaries cannot be attributed to effects of noise (see also *Appendix 1—figure 7*).

So far, we have focused on comparing state boundary time series across regions or between brain regions and events. However, that approach does not allow us to fully understand the different ways in which boundaries can be shared across parts of the cortical hierarchy at specific points in time. To investigate this, we can group timepoints together based on the similarity of their boundary profiles; that is, which searchlights do or do not have a neural state boundary at the same timepoint. We used a weighted stochastic block model (WSBM) to identify groups of timepoints, which we will refer to as 'communities.' We found an optimal number of four communities (see *Figure 7*). These communities group together timepoints that vary in how the degree to their neural state boundaries are shared across the cortical hierarchy: timepoints in the first community show the most widely spread neural state boundaries across the hierarchy, while timepoints in the later communities show less widespread state transitions. We found that from community 1–4, the prevalence of state boundaries decreased for all networks, but most strongly for the FPCN and CON, sDMN, aDMN, and auditory networks. However, the same effect was also seen in the higher visual and SMN and motor networks. This might suggest that boundaries that are observed widely across lower-level networks are more likely to traverse the cortical hierarchy.

We also found a similar drop in prevalence of event boundaries across communities, supporting our previous observation that the perception of event boundaries is associated with the sharing of neural state boundaries across large parts of the cortical hierarchy. We repeated this analysis in two independent groups of participants to be able to assess the stability of this pattern of results. Although group 1 showed an optimum of four communities and group 2 an optimum of five communities, the pattern of results was highly similar across both groups (see *Appendix 1—figure 8*).

## Discussion

While event segmentation is a critical aspect of our ongoing experience, the neural mechanisms that underlie this ability are not yet clear. The aim of this article was to investigate the cortical organization of neural states that may underlie our experience of distinct events. By combining an innovative data-driven state segmentation method with a movie dataset of many participants, we were able to identify neural states across the entire cortical hierarchy for the first time. We observed particularly fast states in primary sensory regions and long periods of information integration in the left middle frontal gyrus and medial prefrontal cortex. Across the entire cortical hierarchy, we observed associations between neural state and perceived event boundaries and our findings demonstrate that neural state boundaries are shared within long-range functional networks as well as across the temporal hierarchy between distinct functional networks.

### A partially nested cortical hierarchy of neural states

Previous findings have suggested that neural states may be organized in a nested cortical hierarchy (*Baldassano et al., 2017*). In line with this hypothesis, we observed that neural state boundaries throughout the entire hierarchy overlap with perceived event boundaries, but this overlap is particularly strong for transmodal regions such as the dorsal medial prefrontal cortex, anterior cingulate cortex, left superior and middle frontal gyrus, and anterior insula. In line with EST, the strong alignment in the anterior cingulate cortex suggests that the disparity between predicted and perceived sensory input may play a role in the experience of an event boundary (*Holroyd and Coles, 2002*; *Kurby and Zacks, 2008*), while the involvement of the dorsal medial prefrontal cortex is in line with previous studies linking this region to representations of specific events (*Baldassano et al., 2018*; *Krueger et al., 2009*; *Liu et al., 2022*).

Once we accounted for the maximal possible overlap given the number of neural states in a particular brain region, we also found strong overlap in unimodal areas such as the visual, auditory, and somatosensory cortices. This finding suggests that some of the neural state boundaries that can be identified in early sensory regions are also consciously experienced as an event boundary. Potentially

because these boundaries are propagated to regions further up in the cortical hierarchy. Which of the boundaries in lower-level areas propagate to higher-order cortical areas may be moderated by attentional mechanisms, which are known to alter cortical information processing through long-range signals (*Buschman and Miller, 2007*; *Gregoriou et al., 2009*). When participants are not attending the sensory input (i.e., during daydreaming), there may be much lower correspondence between neural state boundaries in higher- and lower-level regions.

So, what do these neural states represent? Recent work by *Chien and Honey, 2020* has shown that neural activity around an artificially introduced event boundary can be effectively modeled by ongoing information integration, which is reset by a gating mechanism, very much in line with the mechanism proposed to underlie event segmentation (*Kurby and Zacks, 2008*). Similarly, neural states may represent information integration about a particular stable feature of the environment, which is reset when that feature undergoes a substantial change (*Bromis et al., 2022*). This suggests that neural states in early visual cortex may represent short-lived visual features of the external environment, while states in anterior temporal cortex may contain high-level semantic representations related to the ongoing narrative (*Clarke and Tyler, 2015*). For transmodal regions such as the medial prefrontal cortex, or middle frontal gyrus, that have been associated with many different high level cognitive processes (*Duncan, 2010*; *van Kesteren et al., 2012*; *Simony et al., 2016*), it is not yet clear what a distinct neural state might represent. Just as perceived event boundaries can be related to changes in one or multiple situational dimensions, such as changes in goals or locations (*Clewett et al., 2019*; *Zacks et al., 2009*), neural state boundaries in transmodal cortical areas may not necessarily reflect one particular type of change. State boundaries in these regions are likely also dependent on the goals of the viewer (*Wen et al., 2020*).

We also investigated the factors that distinguish neural state boundaries that traverse the hierarchy from those that do not. It has previously been shown that changes across multiple aspects of the narrative are more likely to result in an experienced event boundary (*Zacks et al., 2010*). In line with this, we observed that boundaries that were represented in more brain regions at the same time were also more likely to be associated with the experience of an event boundary. The strength of the neural state boundary, as measured by the amount of change in neural activity patterns, was also identified as a factor that can to some degree distinguish neural states that appear in subjective experience from the neural states that do not, particularly in temporal cortex, inferior frontal gyrus, precuneus, and medial prefrontal gyrus. This suggests that a neural state boundary is not an all or none occurrence. Instead, the reset of representations at neural state boundaries (*Chien and Honey, 2020*) may differ based on what is happening in other brain regions, on the current attentional focus, or based on the degree of change in the representations of the environment in that particular brain region.

More evidence for the idea of a nested cortical hierarchy of neural state boundaries comes from our connectivity analyses, which show that neural state boundaries are shared both within and across groups of regions that partly resemble well-known functional brain networks. This sharing of boundaries across different cortical areas may suggest that neural states in higher-level cortical regions represent an overarching representation that corresponds to many distinct states in lower-level cortical areas, which all represent different features of that overarching representation (e.g., words spoken, characters on screen, or locations within a particular situation). This is in line with previous conceptualizations of events as partonomic hierarchies (*Zacks et al., 2001a*) and with other models of hierarchical neural representations, such as the hub-and-spokes model for semantic representations, which proposes that semantic knowledge is represented by the interaction between modality-specific brain regions and a transmodal semantic representational hub in the anterior temporal lobe (*Lambon Ralph et al., 2010*; *Rogers et al., 2004*). It is also in line with a recently proposed hierarchical representation of episodic memories, in which items that are linked within small-scale events are in turn linked within large-scale episodic narratives (*Andermane et al., 2021*).

## Timescales of information processing across the cortex

While previous studies have been able to show regional differences in the timescale of information processing across parts of the cortex (*Baldassano et al., 2017*; *Hasson et al., 2008*; *Honey et al., 2012*; *Lerner et al., 2011*; *Stephens et al., 2013*), here we were able to reveal neural state timescales across the entire cortex for the first time. The validity of our results is supported by extensive validations using simulations (see 'Supplementary methods' in Appendix 1 and *Geerligs et al., 2021*)

and the reliability of our observations across independent groups of participants. It is also supported by the similarity between our results and previous findings based on very different approaches, such as experiments with movies and auditory narratives that have been scrambled at different timescales (*Hasson et al., 2008*; *Honey et al., 2012*; *Lerner et al., 2011*), or resting-state fluctuations in electro-corticography (*Honey et al., 2012*) and fMRI data (*Stephens et al., 2013*).

Although we characterized brain areas based on their median state length, we observed that neural states within a region were not of equal duration, suggesting that regional timescales may change dynamically based on the features of the stimulus. This is also in line with the observed correspondence between neural state and perceived event boundaries. Event boundaries have previously been shown to align with changes in features of the narrative, such as characters, causes, goals, and spatial locations (*Zacks et al., 2009*). Therefore, the overlap between state boundaries and perceived event boundaries across the cortex also suggests that characteristics of the sensory input are driving the occurrence of neural state boundaries. Together, these findings show that the timescale of information processing in particular brain regions is not only driven by stable differences in the rate of temporal integration of information, which may be associated with interregional interactions in the neural circuitry (*Honey et al., 2012*), but also by the properties of the input that is received from the environment. Our results show that some of the areas that were not covered in previous investigations (*Baldassano et al., 2017*; *Hasson et al., 2008*; *Honey et al., 2012*; *Lerner et al., 2011*; *Stephens et al., 2013*), such as the medial prefrontal cortex and middle frontal gyrus, have the longest timescales of information processing. This suggests these regions at the top of the cortical hierarchy (*Clarke and Tyler, 2015*; *Fuster, 2001*) also have the slowest timescales of information processing, in line with expectations based on the hierarchical process memory framework (*Hasson et al., 2015*).

## Functional networks of neural state boundaries

In line with previous work (*Baldassano et al., 2017*), we found that neural state boundaries are shared across brain regions. Our results show for the first time that these boundaries are shared within distinct functional networks. Interestingly, the networks we identify partially resemble the functional networks that are typically found using regular functional connectivity analyses (c.f. *Power et al., 2011*; *Yeo et al., 2011*), though there are some differences. For instance, the visual network was segregated into two smaller subnetworks, and for other networks, the topographies sometimes deviated somewhat from those observed in prior work.

Our results show that functional networks defined by state boundaries differ in their timescales of information processing. While some networks have a particular temporal mode of information processing, other networks show a within-network gradient of neural state timescales. For the DMN, we observed a split into posterior, superior, and anterior subnetworks with markedly different timescales. The anterior and posterior subnetworks closely resemble previously observed posterior and anterior DMN subnetworks (*Andrews-Hanna et al., 2010*; *Campbell et al., 2013*; *Lei et al., 2014*), while the superior subnetwork resembles the right dorsal lateral DMN subnetwork (*Gordon et al., 2020*). The posterior/fast DMN is particularly prominent in the precuneus and angular gyri, which are thought to engage in episodic memory retrieval through connectivity with the hippocampal formation (*Andrews-Hanna et al., 2010*). The posterior DMN has also been proposed to be involved in forming mental scenes or situation models (*Ranganath and Ritchey, 2012*). Thus, neural states in this subnetwork may reflect the construction of mental scenes of the movie and/or retrieval of related episodic memories. The superior DMN (or right dorsal lateral DMN) showed timescales of neural states that were in between those of the anterior and posterior DMN. This network has previously been suggested to be a connector hub within the DMN, through which the FPCN exerts top-down control over the DMN. This is in line with the strong state boundary overlap we observed between searchlights in the sDMN and the FPCN. The anterior/slow DMN is particularly prominent in the medial prefrontal cortex that has been related to self-referential thought, affective processing, and integrating current information with prior knowledge (*Benoit et al., 2014*; *Gilboa and Marlatte, 2017*; *van Kesteren et al., 2012*; *Northoff et al., 2006*). The current results suggest that these processes require integration of information over longer timescales.

## Real-life experience

Although event segmentation is thought to be a pivotal aspect of how information is processed in real life (*Zacks et al., 2007*), it is often not considered in experimental settings, where events are predetermined by the trial or block structure. This study and previous work (*Baldassano et al., 2017*) show that we are now able to investigate brain activity as it unfolds over time without asking participants to perform a task. This allows us to study brain function in a way that is much more similar to our daily life experience than typical cognitive neuroscience experiments (*Hamilton and Huth, 2020*; *Lee et al., 2020*; *Willems et al., 2020*). This opens the door for investigations of neural differences during narrative comprehension between groups of participants, such as participants with autism who may have trouble distinguishing events that require them to infer the state of mind of others (*Baron-Cohen, 2000*; *Hasson et al., 2009*), or participants with Alzheimer's disease, who may have trouble with segmenting and encoding events in memory (*Zacks et al., 2006*).

It should be noted that this more naturalistic way of investigating brain activity comes at a cost of reduced experimental control (*Willems et al., 2020*). For example, some of the differences in brain activity that we observe over time may be associated with eye movements. Preparation of eye movements may cause activity changes in the frontal-eye-fields (*Vernet et al., 2014*), while execution of eye movements may alter the input in early sensory regions (*Lu et al., 2016*; *Son et al., 2020*). However, in a related study (*Davis et al., 2021*), we found no age difference in eye movement synchrony while viewing the same movie, despite our previous observation of reduced synchrony with age in several areas (particularly the hippocampus, medial PFC, and FPCN; *Geerligs and Campbell, 2018*), suggesting a disconnect between eye movements and neural activity in higher-order areas. In addition, reducing this potential confound by asking participants to fixate leads to an unnatural mode of information processing, which could arguably bias the results in different ways by requiring participants to perform a double task (monitoring eye movements in addition to watching the movie).

## Conclusion

Here, we demonstrate that event segmentation is underpinned by neural state boundaries that occur in a nested cortical hierarchy. This work also provides the first cortex-wide mapping of timescales of information processing and shows that the DMN fractionates into faster and slower subnetworks. Together, these findings provide new insights into the neural mechanisms that underlie event segmentation, which in turn is a critical component of real-world perception, narrative comprehension, and episodic memory formation. What remains to be addressed is how timescales of different brain regions relate to the types of neural representations that are contained within these regions. For example, does the dissociation between the posterior and anterior DMN reflect relatively fast construction of mental scenes and slow integration with existing knowledge, respectively? Studying brain function from this perspective provides us with a new view on the organizational principles of the human brain.

## Materials and methods
### Participants

This study included data from 265 adults (131 females) who were aged 18–50 (mean age 36.3, SD = 8.6) from the healthy, population-derived cohort tested in stage II of the Cam-CAN project (*Shafto et al., 2014*; *Taylor et al., 2017*). Participants were native English speakers, had normal or corrected-to-normal vision and hearing, and had no neurological disorders (*Shafto et al., 2014*). Ethical approval for the study was obtained from the Cambridgeshire 2 (now East of England – Cambridge Central) Research Ethics Committee. Participants gave written informed consent.

### Movie

Participants watched a black-and-white television drama by Alfred Hitchcock called *Bang! You're Dead* while they were scanned with fMRI. The full 25 min episode was shortened to 8 min, preserving the narrative of the episode (*Shafto et al., 2014*). This shortened version of the movie has been shown to elicit robust brain activity, synchronized across participants (*Campbell et al., 2015*; *Geerligs and Campbell, 2018*). Participants were instructed to watch, listen, and pay attention to the movie.

## fMRI data acquisition

The details of the fMRI data acquisition are described in *Geerligs and Campbell, 2018*. In short, 193 volumes of movie data were acquired with a 32-channel head-coil, using a multi-echo, T2\*-weighted EPI sequence. Each volume contained 32 axial slices (acquired in descending order), with slice thickness of 3.7 mm and interslice gap of 20% (TR = 2470 ms; five echoes [TE = 9.4 ms, 21.2 ms, 33 ms, 45 ms, 57 ms]; flip angle = 78°; FOV = 192 mm × 192 mm; voxel size = 3 mm × 3 mm × 4.44 mm), the acquisition time was 8 min and 13 s. High-resolution (1 mm × 1mm × 1 mm) T1- and T2-weighted images were also acquired.

## Data preprocessing and hyperalignment

The initial steps of data preprocessing for the movie data were the same as in *Geerligs and Campbell, 2018* and are described there in detail. Briefly, the preprocessing steps included deobliquing of each TE, slice time correction, and realignment of each TE to the first TE in the run, using AFNI (version AFNI_17.1.01; https://afni.nimh.nih.gov; *Cox, 1996*). To denoise the data for each participant, we used multi-echo independent component analysis (ME-ICA), which is a very promising method for removal of non-BOLD-like components from the fMRI data, including effects of head motion (*Kundu et al., 2012*; *Kundu et al., 2013*). Co-registration followed by DARTEL intersubject alignment was used to align participants to MNI space using SPM12 software (http://www.fil.ion.ucl.ac.uk/spm).

To optimally align voxels across participants in the movie dataset, we subsequently used whole-brain searchlight hyperalignment as implemented in the PyMVPA toolbox (*Guntupalli et al., 2016*; *Hanke et al., 2009*). Hyperalignment is an important step in the pipeline because the neural state segmentation method relies on group-averaged voxel-level data. Hyperalignment uses Procrustes transformations to derive the optimal rotation parameters that minimize intersubject distances between responses to the same timepoints in the movie. The details of the procedure are identical to those in *Geerligs et al., 2021*. After hyperalignment, the data were highpass-filtered with a cut-off of 0.008 Hz. For the analyses that included 2 or 15 independent groups of participants, we ran hyperalignment separately within each subgroup to make sure that datasets remained fully independent.

## Data-driven detection of neural state boundaries

To identify neural state boundaries in the fMRI data, we used GSBS (*Geerligs et al., 2021*). GSBS performs an iterative search for state boundary locations that optimize the similarity between the average activity patterns in a neural state and the (original) brain activity at each corresponding timepoint. At each iteration of the algorithm, previous boundary locations are fine-tuned by shifting them by 1 TR (earlier or later) if this further improves the fit. To determine the optimal number of boundaries in each brain region, we used the t-distance metric. This metric identifies the optimal number of states, such that timepoints within a state have maximally similar brain activity patterns, while timepoints in consecutive states are maximally dissimilar. The validity of these methods has been tested extensively in previous work, with both simulated and empirical data (*Geerligs et al., 2021*). The input to the GSBS algorithm consists of a set of voxel time courses within a searchlight and a maximum value for the number of states, which we set to 100, roughly corresponding to half the number of TRs in our data (*Geerligs et al., 2021*).

Here we improved on the existing method in three ways to increase the validity and reliability of our results. First, GSBS previously placed one boundary in each iteration. We found that for some brain regions this version of the algorithm showed suboptimal performance. A boundary corresponding to a strong state transition was placed in a relatively late iteration of the GSBS algorithm. This led to a steep increase in the t-distance in this particular iteration, resulting in a solution with more neural state boundaries than might be necessary or optimal (for more details, see the 'Supplementary methods' section in Appendix 1 and , *Appendix 1—figure 1A*). We were able to address this issue by allowing the algorithm to place two boundaries at a time. A 2-D search is performed, which allows the algorithm to determine the location of a new state, rather than identifying a boundary between two states. A restriction to the search is that both boundaries must be placed within a single previously existing state. In some cases, it may be more optimal to place one new boundary than two, for example, when an existing state should be split in two (rather than three) substates. To accommodate this, we allow the algorithm to determine whether one or two boundaries should be placed at a time, based on which of these options results in the highest t-distance.

As a consequence of this change in the fitting procedure, we also adjusted the boundary fine-tuning. While we previously fine-tuned boundaries in the order they were detected (i.e., first to last), we now perform the fine-tuning starting from the weakest boundary and ending with the strongest boundary. These changes to the algorithm are all evaluated extensively in the 'Supplementary methods' section in Appendix 1. Code that implements the improved version of GSBS in Python is available in the State-Segmentation Python package (https://pypi.org/project/statesegmentation/).

The final change compared to our previous work entails the use of deconvolved data. We observed that the algorithm was often unable to differentiate short states from transitions between longer states due to the slow nature of the hemodynamic response. This issue can be resolved by first deconvolving the data. Simulations and empirical results demonstrate that these changes resulted in stark increases in the reliability of our results (see 'Supplementary methods' in Appendix 1). The data were deconvolved using Wiener deconvolution as implemented in the rsHRF toolbox (version 1.5.8), based the canonical hemodynamic response function (HRF; *Wu et al., 2021*). Importantly, we did not use the iterative Wiener filter algorithm as we noticed that this blurred the boundaries between neural states. We also investigated the effects of estimating the HRF shape based on the movie fMRI data instead of using the canonical HRF and found that this did not have a marked impact on the results (see 'Supplementary methods' in Appendix 1).

## Whole-brain search for neural state boundaries

We applied GSBS in a searchlight to the hyperaligned movie data. Spherical searchlights were scanned within the Harvard-Oxford cortical mask with a step size of two voxels and a radius of three voxels (*Desikan et al., 2006*). This resulted in searchlights with an average size of 97 voxels (max: 123; IQR: 82–115); this variation in searchlight size was due to the exclusion of out-of-brain voxels. Only searchlights with more than 15 voxels were included in the analysis.

Previous analyses have shown that neural state boundaries cannot be identified reliably in single-subject data. Instead data should be averaged across a group of at least ~17 participants to eliminate sources of noise from the data (*Geerligs et al., 2021*). As the group size increases, the reliability of the results also increases. Therefore, all the results reported here are with the maximal possible group size. To illustrate the reliability of the cortical hierarchy of state durations and the communities of time-points, we randomly divided the data into two independent samples of ~135 participants each before identifying the optimal number of states. In all other analyses, the figures in the 'Results' section are derived from data with all participants averaged in one big group. Statistical testing to determine statistical significance of these results is done with data in which participants were grouped in 15 smaller independent subgroups of 17/18 randomly selected participants per group.

## Defining event boundaries

Event boundaries in the Cam-CAN movie dataset were identified by *Ben-Yakov and Henson, 2018* based on data from 16 observers. These participants watched the movie outside the scanner and indicated with a keypress when they felt 'one event (meaningful unit) ended and another began.' Participants were not able to rewind the movie. *Ben-Yakov and Henson, 2018* referred to the number of observers that identified a boundary at the same time as the boundary salience. In line with their approach, we only included boundaries identified by at least five observers. This resulted in a total of 19 boundaries separated by 6.5–93.7 s, with a salience varying from 5 to 16 observers (mean = 10).

## Comparison of neural state boundaries to event boundaries

To compare the neural state boundaries across regions to the event boundaries, we computed two overlap metrics; the absolute and relative boundary overlap. Both overlap measures were scaled with respect to the expected number of overlapping boundaries. To compute these values, we define $E$ as the event boundary time series and $S_i$ as the neural state boundary time series for searchlight $i$. These time series contain zeros at each timepoint $t$ when there is no change in state/event and ones at each timepoint when there is a transition to a different state/event.

The overlap between event boundaries and state boundaries in searchlight $i$ is defined as

$$O_i = \sum_{t=1}^{n} E_t \cdot S_{i,t}$$

where $n$ is the number of TRs.

If we assume that there is no association between the occurrence of event boundaries and state boundaries, the expected number of overlapping boundaries is defined as in *Zacks et al., 2001a* as:

$$OE_i = \frac{1}{n} \cdot \sum_{t=1}^{n} E_t \cdot \sum_{t=1}^{n} S_{i,t}$$

Because the number of overlapping boundaries will increase as the number of state boundaries increases, the absolute overlap (*OA*) was scaled such that it was zero when it was equal to the expected overlap and one when all neural state boundaries overlapped with an event boundary. The absolute overlap therefore quantifies the proportion of the neural state boundaries that overlap with an event boundary:

$$OA_i = \frac{O_i - OE_i}{\sum_{t=1}^{n} S_{i,t} - OE_i}.$$

Instead, the relative overlap (*OR*) was scaled such that is was one when all event boundaries overlapped with a neural state (or when all neural state boundaries overlapped with an event boundary if there were fewer state boundaries than event boundaries). In this way, this metric quantifies the overlap without penalizing regions that have more or fewer state boundaries than event boundaries. The relative overlap is defined as

$$OR_i = \frac{O_i - OE_i}{\min\left\{\sum_{t=1}^{n} E_t, \sum_{t=1}^{n} S_{i,t}\right\} - OE_i}.$$

For each searchlight, we tested whether the boundary overlap was significantly different from zero across the 15 independent samples.

In addition to investigating the overlap between the event boundaries and the state boundaries, we also investigated the effect of boundary strength. We define boundary strength as the Pearson correlation distance between the neural activity patterns of consecutive neural states. We investigated whether taking the strength of state boundaries into account improved the absolute overlap compared to using the binary definition of state boundaries. To do this, we change the neural state boundary time series for searchlight $S_i$ such that, instead of ones, it contains the observed state boundary strength when there is a transition to a different state. After redefining $S_i$ in this way, we recomputed the absolute overlap and investigated which brain regions showed a significant increase in overlap when we compare the strength-based absolute overlap ($OA - ST_i$) to the binary absolute overlap ($OA_i$), across the 15 independent samples.

## Quantification of boundary overlap between searchlights

In order to quantify whether the overlap between neural state boundaries between different brain regions was larger than expected based on the number of state boundaries, we computed the relative boundary overlap as described above. We used the relative, instead of the absolute overlap, to make sure that the overlap between regions was not biased by regional differences in the number of states. The relative boundary overlap allows us to quantify the degree to which state boundaries are nested. The overlap between the neural state time series of searchlights $i$ and $j$ is defined as

$$O_{i,j} = \sum_{t=1}^{n} S_{i,t} \cdot S_{j,t}.$$

Here, we used the binary definition of $S_i$ and $S_j$, containing ones when there was a transition between states and zeros when there was no transition.

The expected overlap between neural state boundaries was quantified as

$$OE_{i,j} = \frac{1}{n} \cdot \sum_{t=1}^{n} S_{i,t} \cdot \sum_{t=1}^{n} S_{j,t}.$$

The relative boundary overlap metric (*OR*) was scaled such that it was zero when it was equal to the expected overlap and one when it was equal to the maximal possible overlap:

$$OR_{i,j} = \frac{O_{i,j} - OE_{i,j}}{min\left\{\sum_{t=1}^{n} S_{i,t}, \sum_{t=1}^{n} S_{j,t}\right\} - OE_{i,j}}.$$

For each pair of searchlights, we tested whether the boundary overlap was significantly different from zero across the 15 independent samples.

## Identification of functional networks

In order to identify networks of regions that contained the same neural state boundaries, we computed the boundary overlap between each pair of searchlights as described above. We used the data in which all 265 participants were averaged. Based on the boundary overlap between all searchlight pairs, functional networks were detected using a consensus partitioning algorithm (*Lancichinetti and Fortunato, 2012*), as implemented in the Brain Connectivity Toolbox (*Rubinov and Sporns, 2010*). The aim of the partitioning was to identify networks (groups) of searchlights with high boundary overlap between searchlights within each network and low(er) overlap between searchlights in different networks. Specifically, an initial partition into functional networks was created using the Louvain modularity algorithm (*Blondel et al., 2008*), which was refined using a modularity fine-tuning algorithm (*Sun et al., 2009*) to optimize the modularity. The fit of the partitioning was quantified using an asymmetric measure of modularity that assigns a lower importance to negative weights than positive weights (*Rubinov and Sporns, 2011*).

Because the modularity maximization is stochastic, we repeated the partitioning 100 times. Subsequently, all 100 repetitions for all of the groups were combined into a consensus matrix. Each element in the consensus matrix indicates the proportion of repetitions and groups in which the corresponding two searchlights were assigned to the same network. The consensus matrix was thresholded such that values less than those expected by chance were set to zero (*Bassett et al., 2013*). The values expected by chance were computed by randomly assigning module labels to each searchlight. This thresholded consensus matrix was used as the input for a new partitioning, using the same method described above, until the algorithm converged to a single partition (such that the final consensus matrix consisted only of ones and zeroes).

The procedure described above was applied for different values of the resolution parameter $\gamma$ (varying $\gamma$ between 1 and 3; *Reichardt and Bornholdt, 2006*). Increasing the value of $\gamma$ allows for the detection of smaller networks. We used the same values for $\gamma$ across the initial and consensus partitioning. We selected the partition with the highest similarity to a previous whole brain network partition (*Power et al., 2011*), as measured by aMI (*Xuan Vinh et al., 2010*). We specifically chose the parcellation by *Power et al., 2011* as a reference as it proved functional network labels per voxel, rather than regional of interest, making it more similar to our searchlight analyses. To compare our network labels for each searchlight to the voxelwise Power networks, we labeled each searchlight according to the Power network label that occurred most frequently in the searchlight voxels. The highest similarity was observed for gamma = 1.8 (aMI = 0.39). We named each functional network we identified in accordance with the Power network that it overlapped most with, in addition to a descriptive term about the network location (e.g., ventral, posterior) or function (early, late).

## Co-occurrence of neural state boundaries and events

On the level of functional networks, we investigated the association between neural boundary co-occurrence and event boundaries. Just like our investigation of the role of boundary strength, we investigated whether taking boundary co-occurrence into account would increase the absolute overlap with events. We did this for both boundary co-occurrence within the network that a given searchlight is part of, as well as the co-occurrence across all searchlights in the brain.

To do this, we changed the neural state time series for searchlight $i$ ($S_i$) such that for timepoints with state transitions, it does not contain ones, but the proportion of searchlights within that searchlights' network (or within all searchlights in the brain whole brain) that also show a neural state boundary at that timepoint. After redefining $S_i$ in these ways, we recomputed the absolute overlap. This resulted in three measures of absolute overlap: the binary overlap ($OA_i$), the within-network co-occurrence overlap ($OA - N_i$), and the whole-brain co-occurrence overlap ($OA - WB_i$) and investigated which brain regions showed a significant increase in overlap when we compared $OA - N_i$ and $OA - WB_i$ to $OA_i$, across the 15 independent samples.

To look in more detail at how boundaries that are shared vs. boundaries that are not shared are associated with the occurrence of an event boundary, we performed an additional analysis at the level pairs of searchlights. For each pair of searchlights $i$ and $j$, we created three sets of neural state boundaries time series, boundaries unique to searchlights $i$ or $j$: $S_{i,\sim j}$ and $S_{j,\sim i}$ and boundaries shared between searchlights $i$ and $j$: $S_{i\&j}$. More formally, using the binary definition of the neural state boundary time series $S_i$ and $S_j$, these are defined at each timepoint $t$ as

$$S_{i\&j,t} = S_{i,t} \cdot S_{j,t},$$

$$S_{i,\sim j,t} = S_{i,t} - S_{i\&j,t},$$

$$S_{j,\sim i,t} = S_{j,t} - S_{i\&j,t}.$$

Then, we investigated the absolute overlap between each of these three boundary series and the event boundaries as described in the section 'Comparison of neural state boundaries to event boundaries.' This resulted in three estimates of absolute boundary overlap; for boundaries unique to searchlight $i$ ($OA_{i,\sim j}$) and searchlight $j$ ($OA_{j,\sim i}$) and the shared boundaries ($OA_{i\&j}$). Then we tested whether the absolute overlap for the shared boundaries was larger than the absolute overlap for non-shared boundaries using the searchlight that showed the largest overlap in their unique boundaries as the baseline: $OA_{i\&j} > \max\{OA_{i,\sim j}, OA_{j,\sim i}\}$. Because the absolute boundary overlap is scaled by the total number of neural state boundaries, it is not biased when there is a larger or smaller number of shared/non-shared states between searchlights $i$ and $j$. It is only affected by the proportion of neural state boundaries that overlap with an event boundary. If that proportion is the same for shared and non-shared boundaries, the overlap is also the same.

Finally, we performed an exploratory analysis to further investigate how neural state boundaries are shared across the cortical hierarchy. To this end, we used a WSBM to identify groups of time-points, which we will refer to as 'communities' (*Aicher et al., 2015*). The advantage of WSBM is that it can identify different types of community structures, such as assortative communities (similar to modularity maximization) or core-periphery communities. As the input to the WSBM, we computed the Euclidean distance between the neural state boundary vectors of each timepoint. These neural state boundary vectors contain zeros for searchlights with no boundary and ones for searchlights with a neural state boundary at a specific timepoint. We varied the number of communities from 2 to 10, and we repeated the community detection 1000 times for each number of communities with a random initialization. The WSBM can be informed by the absence or presence of connections and by the connection weights. The alpha parameter $\alpha$ determines the trade-off between the two. Because we used an unthresholded Euclidean distance matrix as the input to the WSBM, we based the community detection only on the weights and not on the absence or presence of certain connections (fixing $\alpha$ to 1). The optimal number of communities was based on the log-likelihood for each number of communities. After identifying the communities, we ordered them based on the average number of neural states per timepoint in each cluster. The algorithm was implemented in MATLAB using code made available at the author's personal website (http://tuvalu.santafe.edu/waaronc/wsbm/).

## Statistical testing and data visualization

The results reported in the article are based on analyses in which data were averaged over all 265 participants (or two groups of ~127 participants for the results in *Figure 1*). To investigate the statistical significance of the associations within each searchlight, network, or network pair, we also ran separate analyses within 15 independent samples of participants. For each metric of interest, we obtained a p-value for each searchlight by testing whether this metric differed significantly from zero across all 15 independent samples using a Wilcoxon signed-rank test (*Wilcoxon, 1945*). p-Values were corrected for multiple comparisons using FDR correction (*Benjamini and Hochberg, 2000*). Results in which the sign of the effect in the one group analysis did not match the sign of the average effect across the 15 independent subgroups were considered non-significant.

For all searchlight-based analyses, p-values from the searchlights were projected to the voxel level and averaged across the searchlights that overlapped each voxel before they were thresholded using the FDR-corrected critical p-value (*Benjamini and Hochberg, 2000*). When projecting the results of the analyses to the voxel level, we excluded voxels for which less than half of the searchlights that covered that voxel were included in the analysis. These excluded searchlights had too few in-brain

voxels (see section 'Whole-brain search for neural state boundaries'). Data were projected to the surface for visualization using the Caret toolbox (*Van Essen et al., 2001*).

## Acknowledgements

LG was supported by a Vidi grant (VI.Vidi.201.150) from the Netherlands Organization for Scientific Research. KC was supported by the Natural Sciences and Engineering Research Council of Canada (grant RGPIN-2017-03804 to KC) and the Canada Research Chairs program. We thank Aya Ben-Yakov for providing data on the perceived event boundaries in the Cam-CAN movie dataset. Data collection and sharing for this project was provided by the Cambridge Centre for Ageing and Neuroscience (Cam-CAN). Cam-CAN funding was provided by the UK Biotechnology and Biological Sciences Research Council (grant number BB/H008217/1), together with support from the UK Medical Research Council and University of Cambridge, UK.

## Additional information

### Funding

| Funder | Grant reference number | Author |
| --- | --- | --- |
| Nederlandse Organisatie voor Wetenschappelijk Onderzoek | VI.Vidi.201.150 | Linda Geerligs |
| Natural Sciences and Engineering Research Council of Canada | RGPIN-2017-03804 | Karen L Campbell |

The funders had no role in study design, data collection and interpretation, or the decision to submit the work for publication.

### Author contributions

Linda Geerligs, Conceptualization, Data curation, Software, Formal analysis, Validation, Investigation, Visualization, Methodology, Writing - original draft, Project administration, Writing - review and editing; Dora Gözükara, Djamari Oetringer, Software, Methodology, Writing - review and editing; Karen L Campbell, Writing - original draft, Writing - review and editing; Marcel van Gerven, Conceptualization, Writing - review and editing; Umut Güçlü, Conceptualization, Software, Methodology, Writing - review and editing

### Author ORCIDs

Linda Geerligs  http://orcid.org/0000-0002-1624-8380
Marcel van Gerven  http://orcid.org/0000-0002-2206-9098

### Ethics

Human subjects: This Cambridge Centre for Ageing Neuroscience study was conducted in compliance with the Helsinki Declaration, and has been approved by the local ethics committee, Cambridgeshire 2 Research Ethics Committee (now East of England - Cambridge Central; reference: 10/H0308/50). Participants gave written informed consent prior to participating in the study.

### Decision letter and Author response

Decision letter https://doi.org/10.7554/eLife.77430.sa1
Author response https://doi.org/10.7554/eLife.77430.sa2

## Additional files

### Supplementary files
• Transparent reporting form

## Data availability

The data used in this project can be requested via - https://camcan-archive.mrc-cbu.cam.ac.uk/data-access/. The code used to generate the results in the paper is available at https://github.com/lgeerligs/NestedHierarchy (copy archived at swh:1:rev:9049f7500c6db1b90b539bcf859e59edb55f5fa6). The improvements to our GSBS algorithm that are presented in this paper are released in a Python package: https://pypi.org/project/statesegmentation/.

The following previously published dataset was used:

| Author(s) | Year | Dataset title | Dataset URL | Database and Identifier |
|---|---|---|---|---|
| Shafto MA, Tyler LK, Dixon M, Taylor JR, Rowe JB, Cusack R, Calder AJ, Marslen-Wilson WD, Duncan J, Dalgleish T, Henson RN, Brayne C, Matthews FE, Cam-CAN | 2014 | Data from the Cambridge Centre for Ageing and Neuroscience | https://camcan-archive.mrc-cbu.cam.ac.uk/dataaccess/ | Cam-CAN Data Portal, Cam-CAN |

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

## Appendix 1

### Supplementary methods

#### Detecting states

Since the publication of our article describing the GSBS algorithm, we discovered some issues that have resulted in several improvements to the algorithm. First, we discovered that for specific brain regions the original GSBS algorithm performed suboptimally; the placement of one new boundary at a late stage in the fitting process resulted in a large increase in the t-distances (our measure of fit; see *Appendix 1—figure 1A*). This suggests that a strong neural state boundary (i.e., demarcating a large change in neural activity patterns) was detected only in a late iteration of the algorithm, which led to an overestimation of the number of neural states. To deal with this problem, we adapted the algorithm, such that it can place two boundaries at the same time, essentially demarcating the location of a new 'substate' within a previously defined state (as described in 'Materials and methods'). In the following, we refer to this adapted version as states-GSBS. This change in the fitting procedure remedied the issues we experienced before (see *Appendix 1—figure 1B*) and resulted in more robust fitting behavior, which we observed across many brain regions. It also resulted in a change in the approach we used to fine-tune boundary locations; while we previously fine-tuned boundary locations based on their order of detection, the order is now determined by the strength of the boundaries (weakest – strongest).

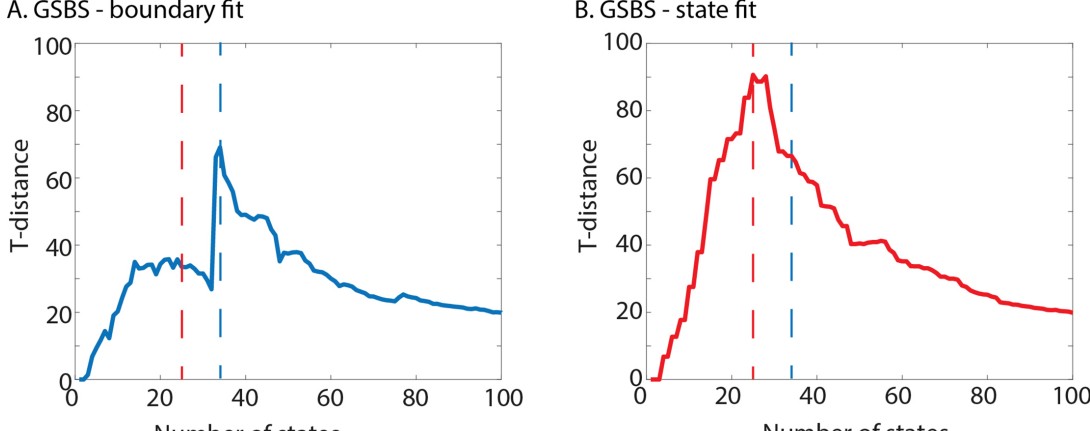

**Appendix 1—figure 1.** T-distance curves and the optimal number of states for the different versions of the greedy state boundary search (GSBS) algorithm. (**A**) The t-distance curve for the original GSBS implementation for an example brain region. (**B**) The t-distance curve for the same brain region with the new option to place two boundaries in one iteration of the algorithm. The dotted lines indicated the optimal number of states for the original GSBS algorithm (blue line) and the states-GSBS algorithm (red line).

To investigate how these changes to GSBS impacted reliability, we split the data in two independent groups of participants and looked at the percentage of overlapping boundaries between the groups for each searchlight. To make sure differences in number of states between methods did not impact our results, we fixed the number of state boundaries to 18 or 19. Because the states-GSBS algorithm can place one or two boundaries at a time, we cannot fix the number of state boundaries exactly, which is why it can be either 18 or 19. We found that the number of overlapping boundaries between groups was substantially higher for states-GSBS compared to the original GSBS implementation and also compared to the GSBS implementation with altered fine-tuning (see *Appendix 1—figure 2A*). This was also the case when we used the optimal number of states as determined by the t-distance, instead of fixing the number of states (see *Appendix 1—figure 2B*). We also investigated the reliability of regional differences in states duration by computing the correlations in median state duration across all searchlights between the two independent groups. Again we observed that reliability increased substantially for states-GSBS compared to the original GSBS implementation and also compared to the GSBS implementation with altered fine-tuning (see *Appendix 1—figure 2C*).

A. Percentage of overlapping boundaries between groups for a fixed number of states (k=18/19)

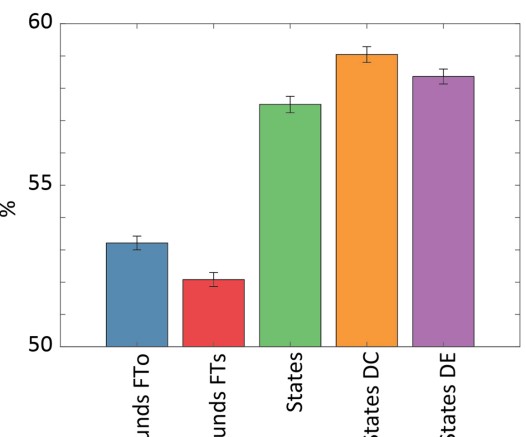

B. Percentage of overlapping boundaries between groups for the optimal number of states

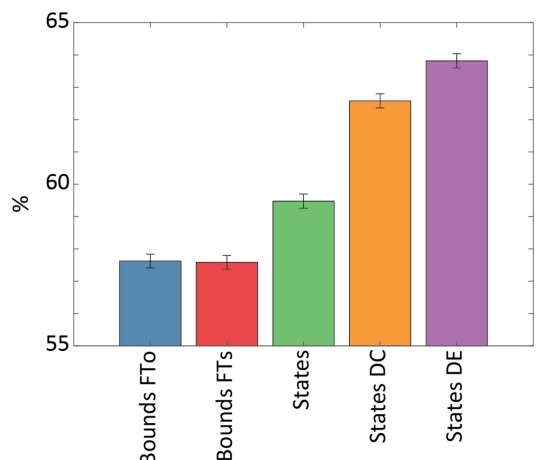

C. Similarity of median state duration across all searchlights between groups

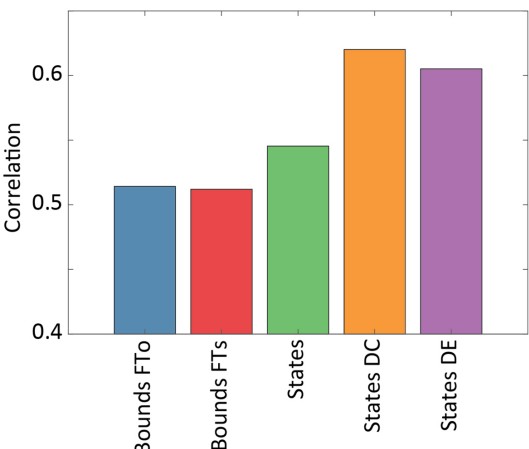

**Appendix 1—figure 2.** Comparing different implementations of the greedy state boundary search (GSBS) algorithm and different data preprocessing steps. (**A**) The percentage of overlapping boundaries between two independent groups for each searchlight. The bar shows the mean across 5029 searchlights, while the error bar shows the standard error. The number of states was fixed to k = 18/19. (**B**) Same as (**A**) but now the number of states was determined by the optimal t-distance. (**C**) The correlation between the estimated median state lengths over all searchlights between two independent groups (correlation computed across 5029 searchlights). Bounds FTo = the original GSBS implementation; bounds FTs = the GSBS implementation with strength-ordered fine-tuning; states = the states-GSBS implementation that can place two boundaries at a time; states DC = states GSBS applied to data deconvolved with a canonical hemodynamic response function (HRF); states DE = states GSBS applied to data deconvolved with an estimated HRF.

## Deconvolution

Another issue we discovered with GSBS is that it was unable to detect short states (of one or two TRs) in some cases. Specifically, we noticed that this happens when consecutive neural states are strongly anticorrelated. We observed such anticorrelated states in many of our searchlights. To investigate this issue, we simulated data with 15, 30, or 50 neural states within 200 TRs using the same setup as our previous work (*Geerligs et al., 2021*). However, instead of randomly generating an activity pattern per state, we used one activity pattern that we inverted when there was a state boundary. This resulted in strongly anticorrelated states (see *Appendix 1—figure 3A*). In this simulated setup, we found that as the number of states increased and there were more states with very short durations, the number of states was underestimated by states-GSBS. We hypothesized that this was

due to the slow hemodynamic response, which obscures transitions between short states. Indeed, when we deconvolved the simulated data, the number of states was estimated correctly, even when there were 50 states (see *Appendix 1—figure 4B*).

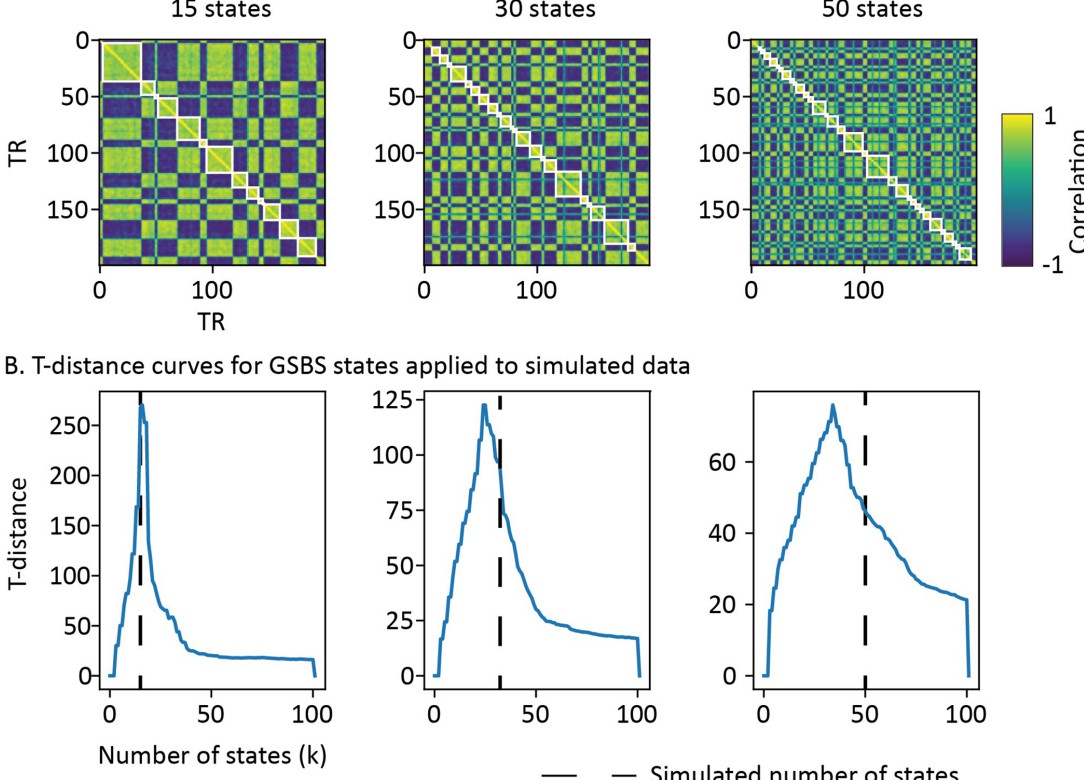

**Appendix 1—figure 3.** Performance of states-GSBS on simulated data with anticorrelated states and (from left to right) 15, 30, or 50 states. (**A**) The correlation matrices with the detected neural state boundaries in white. (**B**) The t-distance curves, where the black line indicates the simulated number of states. For k = 30 and k = 50, the t-distance peaks at a number of states that is below the simulated number of states, suggesting that some of the boundaries of short-lasting neural states are not detected. GSBS, greedy state boundary search.

A. Estimated HRF peak delay

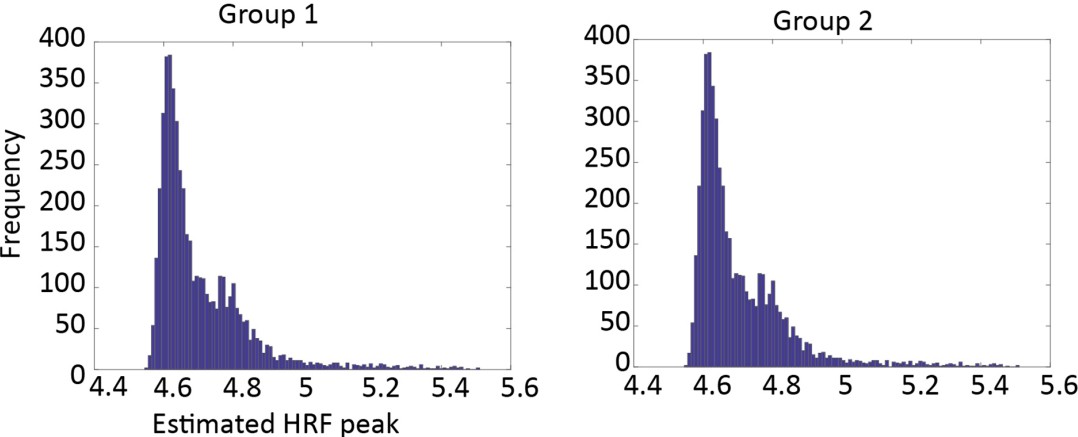

B. Simulated data with different HRF peak delays (k=50)

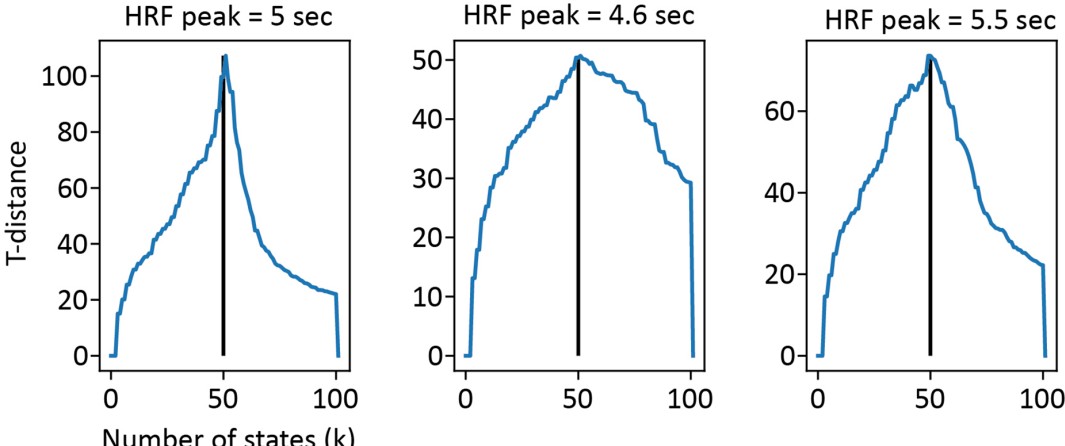

**Appendix 1—figure 4.** Estimated hemodynamic response function (HRF) peak delays and the impact of HRF peak delay on the estimated number of states in simulated data. (**A**) The estimates of the HRF peak delays for all searchlights are shown in a histogram. The values are averaged across all participants within each of the independent groups. The distribution is highly similar for both groups. (**B**) t-distance curves are shown for simulated data with different HRF delays that are deconvolved with a canonical HRF. There is no systematic under- or overestimation of the number of states when the HRF peak delays are in the range of the empirical data.

Because we wanted to be able to identify states with both short and long durations, we chose to apply HRF deconvolution to our data before running states-GSBS using a canonical HRF. We found that deconvolution resulted in a very large improvement in the reliability of regional differences in states duration and also substantially increased the boundary overlap between independent samples (see *Appendix 1—figure 2*). However, it should also be noted that deconvolution may not be optimal for every study interested in neural states. It is particularly important for studies that are interested in accurately identifying the number of neural states in particular brain regions and for studies that are interested in short-lasting states. For studies that are more interested in transitions between longer-lasting states, the deconvolution may actually result in lower signal-to-noise. In particular, because we also observed that deconvolution results in reduced similarity of timepoints within the same state as well as increased similarity of timepoints between states.

## Regional differences in HRF
One concern with deconvolution is that there are known differences between regions in the timing of the HRF (*Taylor et al., 2018*). To investigate whether such differences might impact our results,

we estimated the HRF for each participant and each searchlight using the rsHRF toolbox that is designed to estimate HRFs in resting state data (*Wu et al., 2021*). In this case, we applied the algorithm to our fMRI data recorded during movie watching. Because HRF estimation is applied to single-subject data that contains many sources of noise, we performed some extra data denoising steps (as in *Geerligs and Campbell, 2018*), which included regressing out signals from the CSF and white matter as well as head motion signals. This denoised data was used to run the HRF estimation for each participant and for each voxel within a searchlight. Subsequently, the HRF shape was averaged across all voxels in a searchlight and the data were deconvolved for each participant. Importantly, we used the same data as before (without the extra denoising steps) as the input for the deconvolution to make sure results were comparable. Also, we observed that data cleaning removed some of the signal of interest, resulting in slightly decreased reliability of boundaries. After deconvolution, the data were again averaged within the two independent groups of participants and then we applied the states-GSBS algorithm.

*Appendix 1—figure 4A* shows the estimated HRF peak delays for each brain region after averaging the estimated peaks across all participants within each of the two independent groups. The differences between searchlights in their estimated HRF delay (averaged across participants) were highly reliable across the two independent groups ($r = 0.87$). The peak delays varied between 4.6 and 5.4 s, which is very similar to the delay of the canonical HRF (5 s). When we deconvolved the data with the estimated HRF instead of the canonical HRF, we found that this resulted in a slight decrease in the boundary overlap between independent samples and also slightly reduced the reliability of regional differences in states duration (see *Appendix 1—figure 2*). Regional differences in state duration were highly similar when we compared the deconvolution with the canonical HRF and the deconvolution with the estimated HRF ($r = 0.92$ and $r = 0.93$ for the two groups at the voxel level and $r = 0.73$ and $r = 0.76$ at the level of searchlights). These results suggest that regional differences in the HRF shape did not bias the estimated regional timescales.

To investigate this in more detail, we ran additional simulations to investigate the consequences of slight deviation in the HRF shape on the recovery of the state boundaries (see *Appendix 1—figure 4B*). Simulated data with HRF delays between 4.6 and 5.4 s, which were deconvolved with a canonical HRF, did not show an under- or overestimation in the number of states. Together, these results show that the regional differences we observe in the duration of neural states when we use data that is deconvolved with a canonical HRF cannot be explained by regional differences in the HRF shape. Furthermore, it is not clear that the extra step of estimating the HRF shape results in more accurate or reliable results. That is why we opted for the simpler approach of canonical HRF estimation throughout the article.

## Sample size effects

To look at how replicable results are across samples and how this depends on the sample size, we computed the proportion of boundaries that was shared between each unique pair of participant groups. In line with our previous work, we observed that the boundary time courses were a lot more consistent between different pairs of participant groups when the data were split into two independent groups of 127/128 participants per group (63% of boundaries shared on average) than when the data were split into 15 groups of around 17/18 participants per group (49% of boundaries shared on average). That is why, throughout the article, we report the results with the largest possible sample size. To make sure there were enough unique data points to perform tests for statistical significance to show the consistency of effects across samples, all statistical analyses are performed on the data split into 15 independent groups.

## Supplementary results

### Reliability

As a first step, we determined in which searchlights neural state boundaries were sufficiently reliable for follow-up analyses when we looked at the smallest and therefore least reliable sample size (15 groups with 17/18 participants per group). Specifically, we investigated which searchlights showed a significantly positive Pearson correlation between state boundary time courses in each participant group and the average state boundary time courses across all other participant groups (similar to *Geerligs et al., 2021*). Reliable boundary time courses were observed in 5029 out of 5061 searchlights. The 32 regions without reliable boundary time courses were not included in any of the analyses reported in the article or Appendix 1 ('Supplementary methods' or 'Supplementary results';

see *Appendix 1—figure 5A* for a map of these regions). In these 5029 regions, the reliability was highest for searchlights around the visual and auditory cortex and lowest around the paracentral lobule and the posterior parts of the orbitofrontal cortex (see *Appendix 1—figure 5B*).

## A. Searchlights excluded due to poor reliability

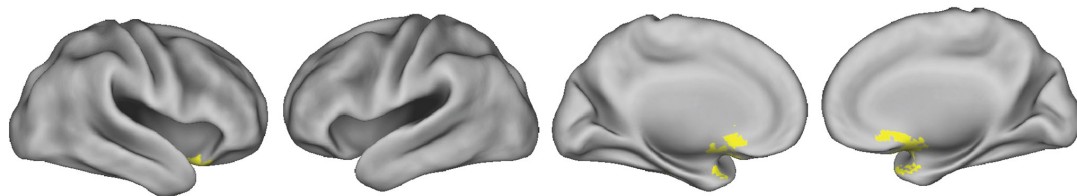

## B. Reliability of neural state boundaries

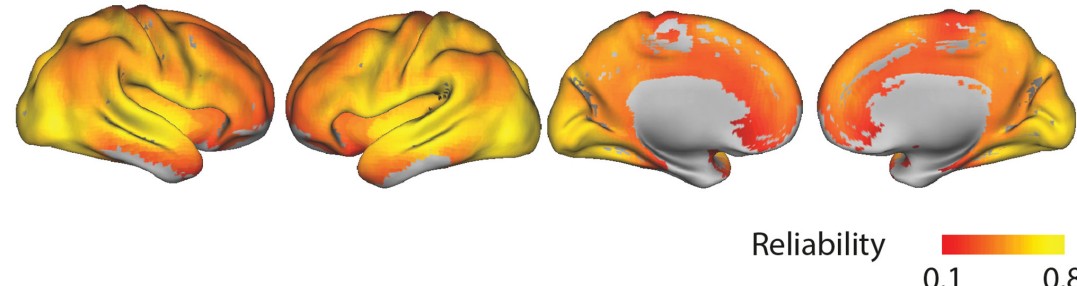

Reliability
0.1    0.8

**Appendix 1—figure 5.** Reliability of neural states boundaries across the brain and an overview of searchlights that were excluded due to poor reliability. (**A**) Searchlights that were excluded due to poor reliability. (**B**) A map of the reliability of neural state boundaries across the cortex.

## Overlap between searchlights

To investigate whether the strong relative boundary overlap between brain regions could be caused by shared sources of noise across brain regions, we recomputed this overlap based on the data from two independent groups of participants. To make sure the resulting matrix remained symmetric, we averaged the results across the two possible orders of participant groups (i.e., comparing searchlight 1 in subgroup 1 to searchlight 2 in subgroup 2, as well as comparing searchlight 1 in subgroup 2 to searchlight 2 in subgroup 1). The results show that the relative boundary overlap computed in this way across independent datasets is highly similar to the boundary overlap within one subgroup ($r$ = 0.69, see *Appendix 1—figure 6A and B*), showing that shared noise cannot be the cause of the strong overlap we observed.

### A. Relative boundary overlap between searchlights

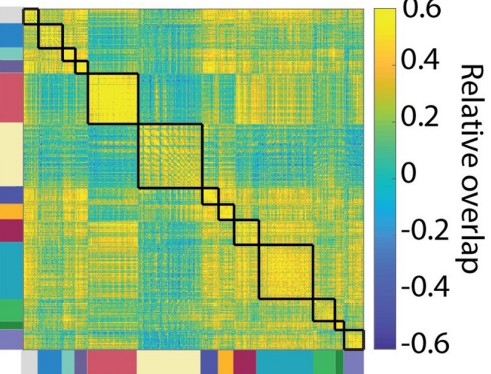

### B. Relative boundary overlap between searchlights across independent groups

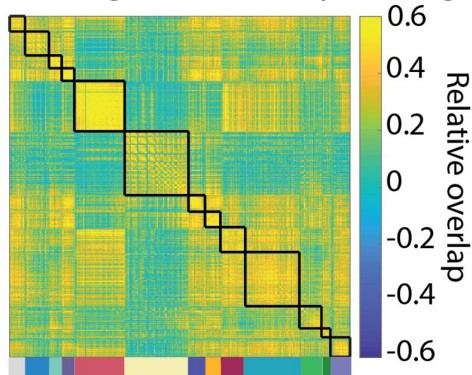

### C. Correlations between average searchlight timeseries

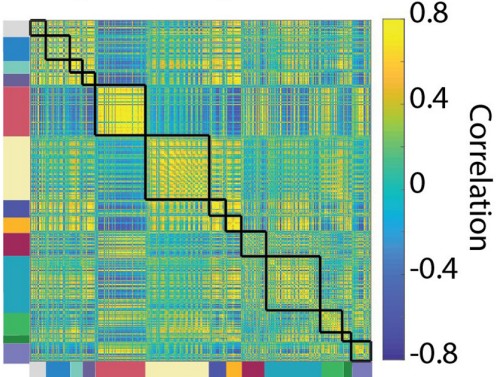

### D. Absolute difference in median state length

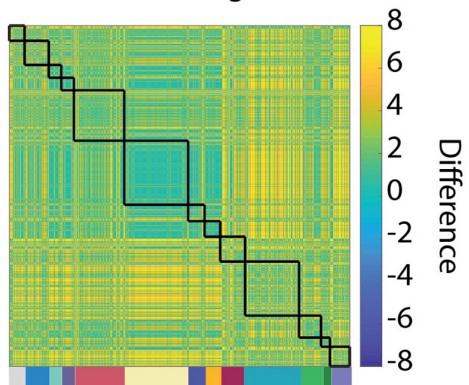

**Appendix 1—figure 6.** Investigating the role of shared noise, ' regular' functional connectivity and regional differences in state length in shaping the boundary overlap between pairs of searchlights. (**A**) The relative neural state boundary overlap between each pair of searchlights. (**B**) Same as (**A**), but computed between two independent groups of participants. This ensures that the overlap cannot be caused by noise shared across brain regions. (**C**) The correlation matrix based on the averaged brain activity time course in each searchlight (i.e., standard measure of functional connectivity). (**D**) The difference between each pair of searchlights in median state length was markedly different from the relative boundary overlap (shown in **A**), showing that the boundary overlap between different regions was not just due to regional differences in the optimal number of states.

To examine whether the relative boundary overlap is simply a proxy for 'regular' functional connectivity, we compared it to the correlation between mean activity time courses in each searchlight (see *Appendix 1—figure 6A and C*). We found that these correlation patterns could only explain a small part of the regional differences in boundary overlap. To investigate whether the boundary overlap is simply a result of regional similarities in state length, we compared the boundary overlap to regional differences in state duration (see *Appendix 1—figure 6A and D*). We found that differences in state duration cannot explain the overlap patterns we observed.

## Effects of noise on overlap between neural states and events for shared boundaries

One concern is that identifying boundaries shared by two regions has a similar effect to averaging, which provides a better estimation of boundaries within each searchlight because it reduces noise. This noise reduction could be the cause of the increased overlap between events and neural states for shared boundaries vs. non-shared boundaries. To investigate this possibility, we examined the increase in overlap for shared vs. non-shared values in the data averaged across 265 participants as well as for each independent subgroup of 17/18 participants. If noise reduction is the cause of

the increase in overlap with event boundaries, we should expect the difference between shared and non-shared boundaries to be largest in the smaller independent subgroups where there is the most to be gained from noise reduction. In contrast, if the increase in overlap with event boundaries is a real effect, not due to noise, its effect size should be larger in the data averaged across all participants, where estimates of boundary locations are more accurate. The results in *Appendix 1— figure 7* show that the latter interpretation is correct, making it unlikely that the observed increase in overlap between neural state and event boundaries is related to noise.

A. Mean increase in absolute overlap for shared vs. non-shared boundaries across all pairs of searchlights

B. Mean increase in absolute overlap for shared vs. non-shared boundaries across pairs of searchlights with a significant increase

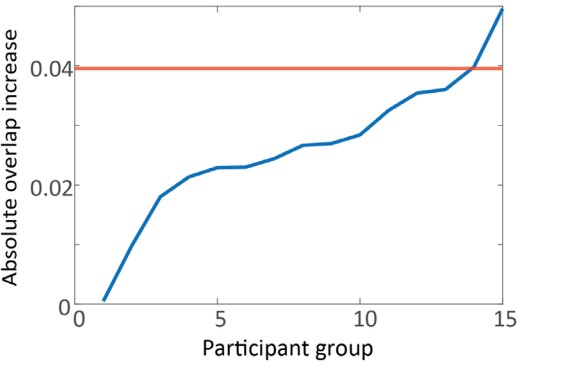
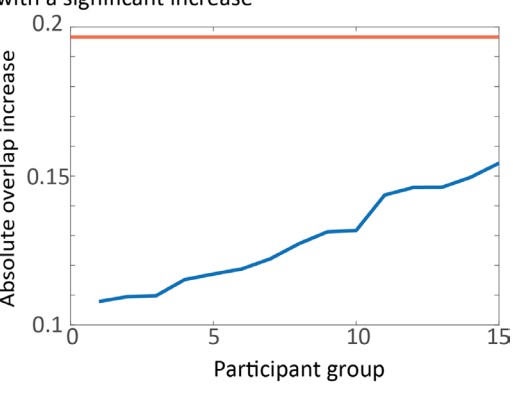

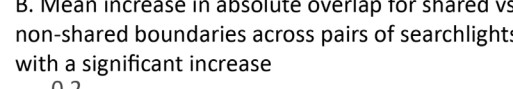

**Appendix 1—figure 7.** Mean increase in absolute overlap for shared vs. non-shared boundaries for large and small groups of participants. (**A**) Mean increase in absolute overlap for shared vs. non-shared boundaries across all pairs of searchlights and (**B**) for the pairs of searchlights that showed a significant increase in overlap. The effect size for the full sample of 265 participants is shown by the red line, and the effect sizes for the independent subgroups of 17/18 participants are shown in blue. The effect size is larger in the data averaged across all 265 participants, suggesting that the increase in overlap is not due to noise reduction.

## Head motion

Another quality check we performed was to investigate the association between neural state boundaries and head motion. First, we computed the average amount of head motion for each TR across all the participants in each of the 15 independent groups. Second, we computed Pearson correlations between the neural state boundary time courses and the average head motion time courses. We investigated whether there was a consistently positive or negative association between state boundaries and head motion across the 15 samples after FDR correlation for multiple comparisons. This was not the case for any of the searchlights.

## Stability of communities of timepoints

In the main text, we identified different communities of timepoints that varied in the degree to which neural state boundaries were shared across the cortical hierarchy. To make sure that these findings are replicable, here we repeated the same analysis across two independent samples of participants (see *Appendix 1—figure 8*). We found that even though the two subgroups did not have the same number of communities (four in group 1, five in group 2), the pattern of results was highly similar across the two. Both groups showed communities that differed in the degree to which boundaries propagated across the cortical hierarchy and in both cases this was also associated with the occurrence of event boundaries, such that timepoints in communities with more widespread boundaries also were more likely to coincide with event boundaries.

## Communities of timepoints for group 1

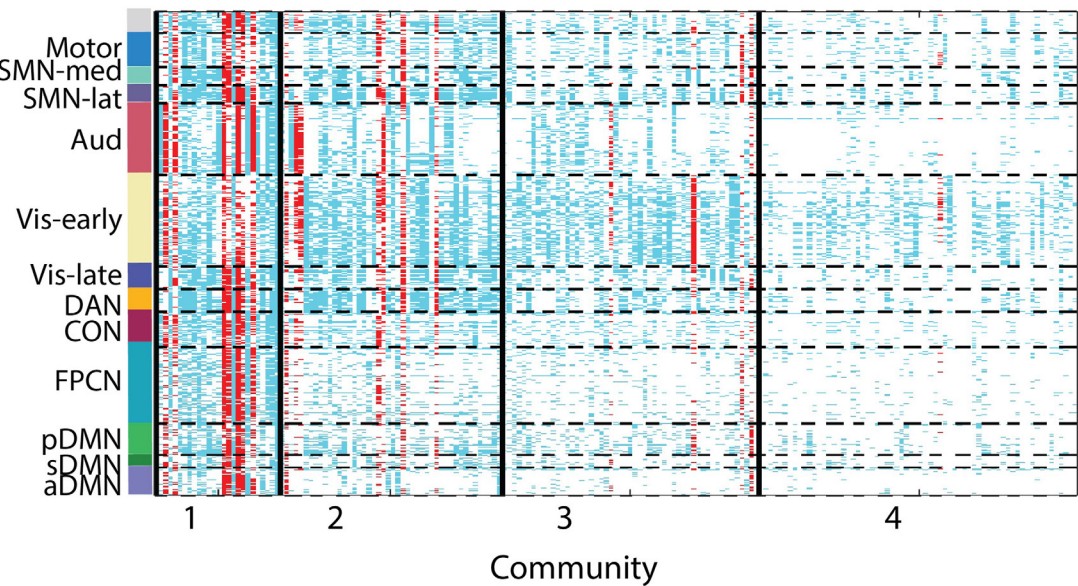

## Communities of timepoints for group 2

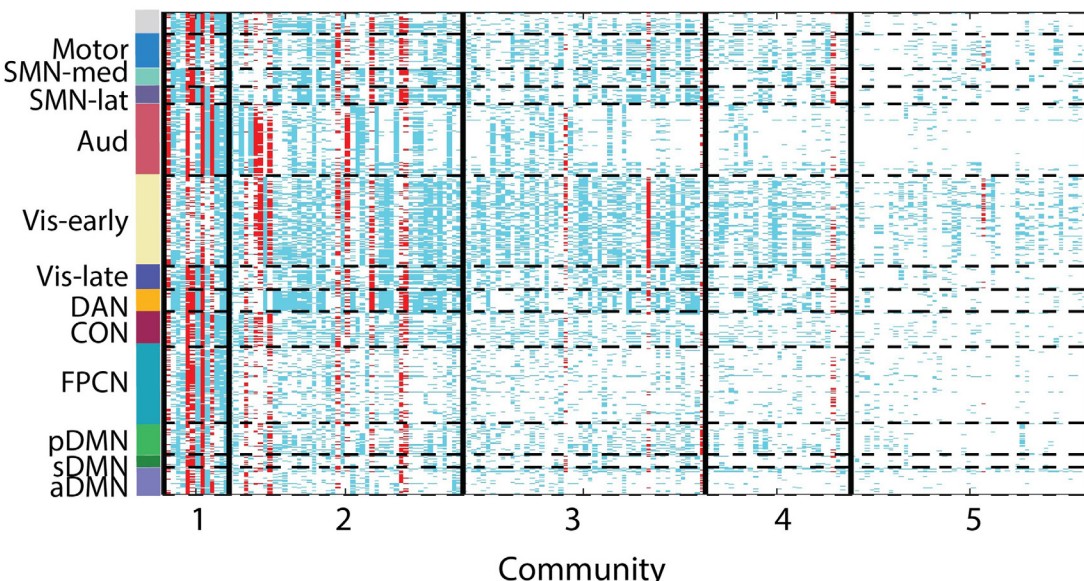

**Appendix 1—figure 8.** Stability of communities of timepoints across two independent groups of participants.

**Appendix 1—table 1.** For each network, the table lists the network defined by *Power et al., 2011* that showed the highest overlap and the percentage of searchlights in the network that overlapped with that particular *Power et al., 2011* network.

| Network name | Power network | Percentage of searchlights |
|---|---|---|
| Motor | Sensorimotor | 49 |
| Sensorimotor-medial | Sensorimotor | 45 |
| Sensorimotor-lateral | Sensorimotor | 37 |
| Auditory | Auditory | 24 |

*Appendix 1—table 1 Continued on next page*

*Appendix 1—table 1 Continued*

| Network name | Power network | Percentage of searchlights |
| --- | --- | --- |
| Visual early | Visual | 58 |
| Visual late | Visual | 20 |
| Dorsal attention network | Dorsal attention network | 27 |
| Cinglulo-opercular network | Cinglulo-opercular network | 23 |
| Fronto-parietal control network | Fronto-parietal task control | 24 |
| Posterior default mode network | Default mode network | 57 |
| Superior default mode network | Default mode network | 24 |
| Anterior default mode network | Default mode network | 68 |

**Appendix 1—table 2.** For the three separate default mode networks (DMNs) we identified, the table lists the overlap with the posterior and anterior DMN defined in *Campbell et al., 2013*.

| Network name | Percentage of searchlights that overlap with anterior DMN | Percentage of searchlights that overlap with posterior DMN |
| --- | --- | --- |
| Posterior DMN | 39 | 70 |
| Superior DMN | 52 | 31 |
| Anterior DMN | 77 | 65 |

