## [Editor Report]

This article addresses the question of how the brain segments naturalistic events and the relationship between perceived event boundaries and neural pattern shifts. By applying an innovative analysis to a large, publicly available dataset, they observe evidence of different timescales of neural state shifts that correspond with perceived event bounds. These results will be of interest to cognitive neuroscientists investigating the relationship between neural states and event segmentation.

---

## [Decision Letter]

**Decision letter after peer review:**

[Editors’ note: the authors submitted for reconsideration following the decision after peer review. What follows is the decision letter after the first round of review.]

Thank you for submitting your work entitled "Timescales and functional organization of event segmentation in the human brain" for consideration by *eLife*. Your article has been reviewed by 3 peer reviewers, and the evaluation has been overseen by a Reviewing Editor and a Senior Editor. The following individuals involved in review of your submission have agreed to reveal their identity: Charan Ranganath (Reviewer #2).

Our decision has been reached after consultation between the reviewers. Based on these discussions and the individual reviews below, we regret to inform you that this submission will not be considered further for publication in *eLife*.

The reviewers were in agreement that this is an important topic and that this research is both interesting and promising. However, the reviewers raised a number of significant concerns that centered around two themes. First, there were a number of points raised about the methodology itself and its validity. After some discussion, it was decided that these methodological points could be addressable through additional analysis and/or simulation, but likely require considerably more work than would be usual for an *eLife* revision. The second set of concerns were with regard to the clear scientific advance over prior work; there was not consensus that these findings move the field forward in a clear way. Reviewer 1 suggested that the generalizability and impact might be improved by drawing direct links to the existing literature, including analysis of a secondary dataset as in the Baldasano et al. (2017) paper. Though, there might be other ways to clarify the impact, as well. Regardless, this is a challenging concern to address in a straightforward way through revision.

As addressing these concerns would likely require more than is typically expected for an *eLife* revision, it was decided to reject this submission. This being said, if you were to undertake the work required to conclusively address these issues, there was sufficient enthusiasm among reviewers that they would be willing to consider this paper again, as a new submission.

I have appended the detailed reviews to this decision letter. I hope you find them constructive with this work.

*Reviewer #1:*

In this paper, Geerligs et al. focus on the alignment of event boundaries across brain regions. They examine the transitions between brain states using the method introduced by Baldassano et al. (2017), and how these state transitions are shared across nodes of large-scale brain networks. They introduce a method that enables them to map event-timescales in a broader set of regions than previously possible, and they use this method to reveal how functional networks of regions share time-aligned "event transitions".

This is a well-written manuscript on a timely and important question.

My main concerns relate to the validity (and potential sources of bias) in the methodology for identifying the event-rate of each region, and I also outline a number of other areas where the conceptual and methodological framing could be improved.

p.3 "This dataset, in combination with the application of hyperalignment to optimize functional alignment (Guntupalli et al., 2016), allowed us to study event segmentation across the entire cortex for the first time, because this dataset shows reliable stimulus-driven activity (i.e., significant inter-subject correlations) over nearly all cortical brain regions (Geerligs et al., 2018). "

A central methodological question, which affects almost every claim in this manuscript, is whether the inference of event boundaries from the HMM model (the methods in Figure 1) is valid, and in what ways it might be biased. The validity question is simple: does it measure what it is supposed to measure? In particular, I would like the authors to justify the final step, in which they compute the difference between the correlation for real boundaries and the correlation for random boundaries. Surely, this difference computation will be affected by the noise ceiling of the individual ROI being examined? I understand why using the random condition as a "reference" makes some sense, but I do not understand why the final decision is made based on the simple arithmetic difference of the mean value for the random boundaries and real boundaries? I suggest that the authors justify this procedure using a simulation procedure where the ground truth about event transitions is known, and the procedure should be compared against the method applied in the original Baldassano et al. (2017) paper.

The bias question is also fairly simple: which factors influence the "k" that is inferred? In particular, if a region has high reliability or low reliability of its response across subjects, does this affect the number of events that will be inferred for that region using the HMM procedure? As noted above, this simulation could additionally investigate how the "k" value varies as function of the noise level (i.e. response reliability) of the ROI.

Additionally, although hyperalignment render a larger swathe of cortex available to analysis, but there will still be variability in the reliability of the signal across regions, and this might interact with the hyperalignment performance. In particular, the accuracy of the hyper alignment procedure (for each subject) will presumably also increase for regions whose reliability of response is higher; it is therefore very to consider whether noise (in "space") introduced by the hyperalignment procedure (and varying across regions as a function of their reliability) could further bias the measurement of the event-timescale via the HMM procedure.

Finally, to better understand this method, the authors could also apply their approach to the freely available data from the Baldassano et al. (2017) paper. Does this method produce results that are at least qualitatively similar? This could help to resolve the question of why the event timescales in this paper are shorter than those observed in the Baldassano et al. paper.

p.7: Event networks: "We found that event boundaries are shared within long-range networks that resemble the functional networks that are typically identified based on (resting state) timeseries correlations (see figure 3A)".

This is one of the most intriguing aspects of this paper. However, it would be much more convincing if the authors would replace their qualitative language (e.g. "resemble") with quantitative metrics of overlap. The overlap could be measure between (a) networks defined based on event-timing and (b) networks defined based on functional connectivity. All of the major functional networks should be available in atlases (e.g. the Yeo lab atlases) or via data sharing repositories. Thus, the authors should be able to substantiate their broad claims of "resemblance" with quantitative demonstrations of how well the event-networks match the functional-connectivity-networks. All of the visual networks as well as the FPN and DMN should be quantitatively compared against standard networks defined elsewhere in the literature.

On the same point: p.13 "The fractionation of the DMN into a fast and slow subnetwork closely aligns with the previously observed posterior and anterior DMN subnetworks (Andrews-Hanna et al., 2010; Campbell et al., 2013; Lei et al., 2014)."

Again, please quantify the alignment when claiming spatial alignment with prior findings.

p.13 "Our results show for the first time that neural events are shared across brain regions in distinct functional networks. "

The authors should consider re-wording this sentence to distinguish their findings from what was already shown in Figure 4B of Baldassano et al. (2017). In particular, note the commonality of event boundaries across early visual and late visual areas (part of the visual network), as well as the commonality of events across angular gyrus and posterior medial cortex (parts of the DMN).

On a related note, in the Abstract we read: "This work extends the definition of functional networks to the temporal domain" – I am unclear on how novel this extension is. To the best of my understanding, the concept of dynamic functional connectivity is not new (e.g. Hutchison et al., 2013), and even second-order pattern-transition methods have been employed to study functional networks (e.g. Anzellotti and Coutanche, 2018). I would like the authors to sharpen their argument for why this result is not entirely expected in light of prior work. Shouldn't members of the same functional networks be expected to exhibit state-transitions at rates higher than chance?

p.11. I struggled to follow the logic of the analysis employed in Figure 6. Why is event duration being predicted from individual frequency bands of the PSD? There is voluminous evidence for band-specific and region-specific artifact (e.g. Birn et al., 2013; Shmueli et al., 2007). Furthermore, distinct functional networks have distinct frequency profiles and coherence patterns (e.g. Salvador et al., 2008; Baria et al., 2011; Stephens et al., 2013). Finally, the frequency bands in the PSD are non-independent (because of the temporal smoothing in the BOLD signal). Therefore, the relationship between frequency band and event duration is confounded by (i) non-independence of frequencies and (ii) frequency covariation across brain regions which arises for a multitude of reasons. The results in Figure 6A seem rather noisy to me, and I imagine that this is because the regression procedure on the PSD is influenced by many interacting and confounding variables.

Another region why this analysis produces (in my opinion) curious results is that it spans distinct sensory modalities which are already known to have opposite PSD-event relationships: along the auditory pathway, PSDs get flatter as event time-scales get longer, while in the visual pathway, PSDs in V1 are already very steep, even while the event timescales are short. It is not clear what is gained by fitting a single model to regions with obviously different relationships of PSD and event structure.

p.12. "These results suggest that visual and auditory stimulation are a prerequisite for observing the temporal hierarchy we describe in this paper and that this hierarchy only partly reflects an intrinsic property of brain function that is also present in the resting state."

I do not follow the logic supporting this claim. How can we know whether the (event-based) temporal hierarchy is preserved in the resting state unless we can measure the event transitions in the resting state data? Isn't this analysis just another way of saying that the PSDs have different shapes during rest and during movie viewing?

References

Anzellotti, S., and Coutanche, M. N. (2018). Beyond functional connectivity: investigating networks of multivariate representations. Trends in cognitive sciences, 22(3), 258-269.

Baria, A. T., Baliki, M. N., Parrish, T., and Apkarian, A. V. (2011). Anatomical and Functional Assemblies of Brain BOLD Oscillations. Journal of Neuroscience, 31(21), 7910-7919. https://doi.org/10.1523/JNEUROSCI.1296-11.2011

Birn, R. M., Diamond, J. B., Smith, M. A., and Bandettini, P. A. (2006). Separating respiratory-variation-related fluctuations from neuronal-activity-related fluctuations in fMRI. Neuroimage, 31, 1536-1548. https://doi.org/10.1016/j.neuroimage.2006.02.048

Coutanche, M. N., and Thompson-Schill, S. L. (2013). Informational connectivity: identifying synchronized discriminability of multi-voxel patterns across the brain. Frontiers in human neuroscience, 7, 15.

Hutchison, R. M., Womelsdorf, T., Allen, E. A., Bandettini, P. A., Calhoun, V. D., Corbetta, M.,.… Chang, C. (2013). Dynamic functional connectivity: Promise, issues, and interpretations. NeuroImage, 80, 360-378. https://doi.org/10.1016/j.neuroimage.2013.05.079

Salvador, R., Martínez, A., Pomarol-Clotet, E., Gomar, J., Vila, F., Sarró, S.,.… Bullmore, E. (2008). A simple view of the brain through a frequency-specific functional connectivity measure. NeuroImage, 39(1), 279-289. https://doi.org/10.1016/j.neuroimage.2007.08.018

Shmueli, K., van Gelderen, P., de Zwart, J. A., Horovitz, S. G., Fukunaga, M., Jansma, J. M., and Duyn, J. H. (2007). Low-frequency fluctuations in the cardiac rate as a source of variance in the resting-state fMRI BOLD signal. Neuroimage, 38(2), 306-320.

Stephens, G. J., Honey, C. J., and Hasson, U. (2013). A place for time: The spatiotemporal structure of neural dynamics during natural audition. Journal of Neurophysiology, 110(9), 2019-2026. https://doi.org/10.1152/jn.00268.2013

*Reviewer #2:*

In this paper, Geerlings and colleagues leverage a large, publicly-available dataset in order to assess shared and distinct timescales of neural pattern shifts at event boundaries across different areas of the brain. In line with prior work, the authors report a gradient of timescales in neural event segmentation, with sensory regions comprising the fastest-shifting areas and 'default mode' nodes such as precuneus and medial prefrontal cortex comprising the slowest-shifiting areas. Importantly, the authors build on this previous research and demonstrate that canonical functional networks – such as the frontoparietal network, and the 'default mode' network – feature distinct subnetworks with corresponding faster and slower timescales of pattern shifts. Finally, a fairly novel analysis applied to these types of data examined power spectral density across regions, which could be used to predict event duration across regions (consistent with observed pattern shifts), and could partly, but not entirely, characterize resting-state fMRI data (suggesting that the audiovisual stimulus drove additional functional properties in brain networks not observed during rest).

Overall, this is an interesting and timely study. The question of how the brain segments naturalistic events is one of increasing popularity, and this manuscript approaches the question with a large sample size and fairly thorough analyses. That said, there are a number of questions and concerns, primarily regarding the analyses.

• Procedures such as hyperalignment, or the related shared response model used by Baldassano and colleagues, are typically implemented by training on one set of the data, and applying the alignment procedure to a separate, held-out dataset (i.e., training and testing sets). It is unclear whether this approach was taken in the current study, or whether the hyperalignment algorithm was trained and tested on same dataset. In the latter case, there is a degree of circularity in the way across-participant alignment was conducted, potentially leading to biased correlation measures. The movie used in the CamCAN dataset is only 8 minutes long, which is probably not enough data for obtaining separate training and test datasets. However, this is still potentially a serious issue for this manuscript, and I am not sure if the use of hyperalignment is appropriate. If I have misunderstood the methodology, it perhaps warrants some clarification in how the training and application of the hyperalignment algorithm proceeded. (I will note that I am aware you used cross-validation for deriving the number of events, but that is unfortunately a separate issue from a train-test split in the hyperalignment routine itself.)

• A key finding from the study is that the FPN and DMN fractionate into different subnetworks that have fast and slow timescales. As noted above, the present results are based on an analysis of data from a relatively short period of time. Although the sample size is very large, one wonders whether this distinction would remain solid with a longer movie. With a very short movie, one can only sample a small number of real events, and this could lead to some instability in estimates of the timescale of representations in relation to the events. This might be an issue in relation to the differentiation of fast and slow subnetworks within the FPN and DMN. For instance, Figure 3B, suggests that the fit values for the slow FPN remain more or less stable across a range of event durations (which presumably reflect k values?). The slow FPN shows an interesting bimodal distribution (as do many of the networks) with the second peak coinciding with the peak for the fast FPN. The differentiation is a bit more convincing for the fast and slow DMN, but it is still not clear whether there are enough events and enough fMRI data from each subject to ensure reliable estimates of the timescales. Just to provide some context for this point, some estimates suggest that reliable identification of resting state networks requires at least 20 minutes of fMRI data.

• Throughout the paper, fMRI results are described in reference to event processing, but the relationship is underdeveloped. Much of the paper relies on the Hidden Markov Model, which assumes that there is a pattern that remains stationary throughout an event. Baldassano's data shows a surprisingly strong correspondence in posterior medial cortex, but it is less clear whether this assumption is valid for other areas. In relation to this point, one can think of event processing as an accumulation of evidence. At the onset of an event, one might have a decent idea of what is about to happen, but as information comes in, the event model can be refined to make stronger predictions. These kinds of within-event dynamics would be lost in the Hidden Markov model. A related point is that the paper conflates timescales of neural states with psychologically meaningful conceptions of events. EST suggests that event segmentation is driven by prediction error-by one interpretation of the model, sensory information can change considerably without leading one to infer an event boundary. However, change in incoming sensory information would almost certainly lead to the detection of "event boundaries" across short timescales in sensory cortical areas. Figure 5 makes it fairly clear that there is a pretty strong distinction to be made between data-driven event identification based on the fMRI data and psychologically meaningful events inferred by the subjects. It would be helpful for the authors to be more clear about what the data do and do not show in relation to putative event cognition processes.

• Why were voxels with an intersubject correlation of less than r=0.35 excluded from analyses? Is this based on prior studies or preliminary analyses? It is not necessarily a bad thing if this choice was made arbitrarily, but I imagine this threshold could have important impacts on the data as presented, so it is worth clarifying.

• Was ME-ICA the only step taken to account for head motion artifacts? If so, there is some concern about whether this step was sufficient to deal with the potential confound. This is especially critical given the fairly brief time series being analyzed here. It would be more compelling to see a quantitative demonstration that head motion is not correlated with the measures of interest.

• A related issue is that of eye movements. Eye movements are related to event processing (e.g., Eisenberg et a., 2018), so one can expect neural activity related to event prediction/prediction error to be confounded with lower-level effects related to eye movements. For instance, we might expect signal artifacts in the EPI data, as well as neural activity related to the generation of eye movements, and changes in visual cortex activity resulting from eye movements. It is unlikely that this issue can be conclusively addressed with the current dataset, and it's not a deal-breaker in the sense that eye movements are intrinsically related to naturalistic event processing. However, it would be useful for the authors to discuss whether this issue is a potential limitation.

• The power spectral analyses were a bit difficult to follow, but more importantly, the motivation for the analysis was not clearly described. The main take home points from this analyses are nicely summarized at the end of p. 14, but it would be helpful to clarify the motivation for this analysis (and the need for doing it) on p.11 in the Results section. Relatedly, is Figure 6A an example spectrum from a particular voxel or region, or an average across regions?

• The take-home message appears to be that different brain networks have different timescales at which they seem to maintain event representations. Moreover, certain networks (e.g., the posterior medial/'default mode' network) do not have uniformly fast or slow timescales. The network-based analysis used here is indeed novel, but the impact of the work could be enhanced by clarifying the significance of the results in relation to what we know about event processing. The explicit demarcation of 'fast' and 'slow' subnetworks may be the key conceptual advance, as was the power spectral analysis, but it isn't clear whether these conclusions could also be ascertained from the maps shown in Baldassano et al., 2017 or other papers from the Hasson group.

This review was completed by Zach Reagh, Ph.D. in collaboration with Charan Ranganath, Ph.D. (I sign all reviews)

*Reviewer #3:*

Geerligs and colleagues conduct a thorough set of analyses aimed at identifying event segmentation timescales across the cortex in a large cohort of participants. They extend previous work by Baldassano et al. by covering the entire cortex, and nicely control for the power spectrum of different regions. In addition, they examine which regions share the same event boundaries, not just the same timescale, and relate these to functional connectivity networks. Overall, their work is impressive and rigorous, but there are a few points that make it somewhat difficult to assess the how strong the contribution is to our understanding of processing timescales:

1. The authors divide the brain into functional networks based on boundary similarity and find that this division is very similar to functional networks defined using resting-state timeseries correlations. They further find increased similarity between regions of different networks that are that are interconnected. Wouldn't the similarity between boundary vectors be strongly linked to the timeseries correlations (both between regions in the same network and across networks)? While the similarity-based functional networks aren't completely identical to those identified in rest, perhaps the same results would be obtained by correlating timeseries in this specific dataset, using the movie data (altering the interpretation of the results).

2. It seems that the power spectrum analysis is run both on the resting-state data and on the movie data, whereas the timescale segmentation is run only on the movie data. I expect this is because hyperalignment is possible only when using a shared stimulus, and the HMM is run only on the hyperaligned data. However, this may bias the correlations presented in figure 6 – the movie PSD-based timescale estimation would be expected to be more similar to the HMM timescales than the rest, simply because the same data is used. A more convincing analysis would be to run the HMM on the rest data as well, and test for correlations between the two estimations of event timescales in the rest data, although this would entail substantial additional analyses (as HMM would also have to be run on non-hyperaligned movie data for comparability). It would also help with point 1, testing whether similarity in boundary vectors arises directly from timeseries correlations. I realize this adds quite a bit of analysis, and the authors may prefer to avoid doing so, but the conclusions arising from the power spectrum analysis should be softened in the Results and Discussion, clearly mentioning this caveat.

3. It would aid clarity to better separate the current contributions from previous findings, in the Results, and mainly in the Discussion. The authors do describe what has previously been found, citing all relevant literature, but it would be helpful to have a clear division of previous findings and novel ones. For example in the first paragraph of the Discussion, and in general when discussing the interpretation of activity the different regions (currently regions that have already been found are somewhat intermixed with the new regions found).

[Editors’ note: further revisions were suggested prior to acceptance, as described below.]

Thank you for resubmitting your work entitled "A partially nested cortical hierarchy of neural states underlies event segmentation in the human brain" for further consideration by *eLife*. Your revised article has been evaluated by Michael Frank (Senior Editor) and a Reviewing Editor.

The manuscript has been improved but there are some remaining issues that need to be addressed, as outlined below:

The reviewers were positive about the revisions you made to this submission and felt that extensive work had been done to improve the paper. There were a few remaining points raised by this review that could be addressed the further strengthen the paper. The Reviewing Editor has drafted this to help you prepare a revised submission.

Essential revisions:

1. Reviewer 1 has raised some additional points for clarification in their review, as noted below. These should be clarified in a revision. Please refer to the comments below for these notes.

2. Some of the conclusions do not completely reflect the results. If additional analyses are not added, perhaps these conclusions could be rephrased, such as "some of the neural boundaries are represented throughout the hierarchy.… until eventually reflected in conscious experience" (p. 14) and "boundaries that were represented in more brain regions at the same time were also more likely to be associated with the experience of an event boundary" (p. 15).

3. Since the GSBS algorithm was fine-tuned based on the data that was later used for analysis, it would be helpful to include additional information demonstrating the choices in the optimization procedure are independent of the eventual results. For example, it isn't clear what 'important boundaries being detected late' means, whether that indicates event boundaries were being missed by the original algorithm. Combined with the fact part of the optimization was based on fixing the number of state boundaries to the number of event boundaries – could these choices have increased the chance of finding overlap between state boundaries and event boundaries?

4. Two small notes: the network defined as posterior DMN includes anterior regions, which is slightly confusing; were the regional differences in HRF assessed on the resting state data or the movie watching data?

Additional Suggestions for Revision (for the authors):

One of the reviewers had some suggestions for additional analyses that might strengthen the results. We pass them along to you here, but you should view these as optional. Only include them if you agree that they will strengthen the conclusions.

there are a few analyses that may help strengthen the conclusions - these are suggested as optional additional analyses, but the authors should feel free not to include them:

• To verify the overlap between searchlights is not due to various artifacts, it may be preferable to compare the searchlight in one region with the searchlights of other groups in the second region (following the rationale of intersubject functional connectivity vs. functional connectivity). It would also be interesting to further explore the nature of the overlap - to see whether there are specific state boundaries that drive most of the overlap or whether different pairs of regions have different overlapping boundaries. This could be used to explore the nature of the hierarchy between regions, beyond just finding that higher regions share boundaries with lower regions. For example, it could enable testing whether state shifts shared by multiple lower level regions are the ones that traverse the hierarchy.

• Further to this, it would be interesting to test whether event boundaries and non-event neural state boundaries form a similar hierarchy (though this may not be feasible with such a low number of event boundaries).

• To assess the effects of noise reduction on the overlap between neural state boundaries and event boundaries, it may be worth testing whether neural state boundaries shared across groups of participants are also more likely to be event boundaries (and specifically whether this effect is stronger in the same regions arising from the co-occurrence analysis). This analysis wouldn't provide an answer, but could help shed some light on the role of noise reduction.

*Reviewer #1:*

This work investigates timescales of neural pattern states (periods of time with a relatively stable activity pattern in a region) across the brain and identify links between state shifts and perceived boundaries events. In multiple regions, they find significant overlap between state shifts and event boundaries, and an even stronger overlap for state shifts that occur simultaneously in more than one region. The results are interesting and timely and extend previous work by Baldassano et al. that found a similar hierarchy in a specific set of brain regions (here extended to the entire cortex).

Strengths

The question of whether neural state shifts form a hierarchy such that state shifts in higher regions coincide with state shifts in sensory regions, and the question of whether event boundaries occur at conjunctions of shifts in different regions are both very interesting.

The optimized GSBS method nicely overcomes limitations of previous methods, as well as a previous version of GSBS. In general, justification is provided for the different analysis choices in the manuscript.

The current work goes beyond previous work by extending the analysis to the entire cortex, revealing that state shifts in higher regions of the cortex overlap with state shifts in lower regions of the hierarchy.

Weaknesses

One of the important conclusions of the paper is that simultaneous neural state shifts in multiple brain regions are more likely to be experienced as boundaries. This finding fits in nicely with existing literature, but the analysis supporting it is not as compelling as the rest of the analyses in the paper:

1. The methods section describing the analysis is not entirely clear. Do Oi, Oj refer to the number of neural state boundaries in searchlights I,j? Or the number of neural state boundaries in each that overlap with an event boundary? If the former (which was my initial interpretation), then how is the reference searchlight chosen – max {Oi,Oj}, as indicated by the formula, or the searchlight with the larger overlap of its unique boundaries (and is the overlap calculated in numerical value or the proportion of overlap)? Given the unclarity, it is difficult to assess whether the degree of overlap between neural state boundaries and event boundaries in each of the searchlights (and/or the number of boundaries in each) could affect the results. It would be helpful to provide verification (either mathematically or with simulations) that higher overlap in one/both searchlights does not lead to a larger difference in overlap between shared and non-shared boundaries.

2. The analysis focuses on pairs of searchlights/regions, demonstrating that in a subset of regions there is a higher chance of an overlap with event boundaries for neural state boundaries that are shared between two regions. Yet the interpretation goes beyond this, suggesting that "boundaries that were represented in more brain regions at the same time were also more likely to be associated with the experience of an event boundary". Additional analyses would be needed to back this claim, demonstrating that overlap between a larger number of regions increases the chance of perceiving a boundary.

3. Could the effect be due to reduction of noise rather than event boundaries arising at neural state boundaries shared between regions? Identifying boundaries shared by two regions has a similar effect to averaging, which the authors have indeed found reduces noise and provides a better estimation of boundaries within each searchlight. This possibility should be discussed.

Recommendations for the authors:

1. As this is a revision of a previous version of the manuscript, and the authors have already conducted a great deal of work to address previous concerns, I am hesitant to suggest additional analyses. However, there are a few analyses that may help strengthen the conclusions – these are suggested as optional additional analyses, but the authors should feel free not to include them:

• To verify the overlap between searchlights is not due to various artifacts, it may be preferable to compare the searchlight in one region with the searchlights of other groups in the second region (following the rationale of intersubject functional connectivity vs. functional connectivity). It would also be interesting to further explore the nature of the overlap – to see whether there are specific state boundaries that drive most of the overlap or whether different pairs of regions have different overlapping boundaries. This could be used to explore the nature of the hierarchy between regions, beyond just finding that higher regions share boundaries with lower regions. For example, it could enable testing whether state shifts shared by multiple lower level regions are the ones that traverse the hierarchy.

• Further to this, it would be interesting to test whether event boundaries and non-event neural state boundaries form a similar hierarchy (though this may not be feasible with such a low number of event boundaries).

• To assess the effects of noise reduction on the overlap between neural state boundaries and event boundaries, it may be worth testing whether neural state boundaries shared across groups of participants are also more likely to be event boundaries (and specifically whether this effect is stronger in the same regions arising from the co-occurrence analysis). This analysis wouldn't provide an answer, but could help shed some light on the role of noise reduction

2. Some of the conclusions do not completely reflect the results. If additional analyses are not added, perhaps these conclusions could be rephrased, such as "some of the neural boundaries are represented throughout the hierarchy.… until eventually reflected in conscious experience" (p. 14) and "boundaries that were represented in more brain regions at the same time were also more likely to be associated with the experience of an event boundary" (p. 15).

3. Since the GSBS algorithm was fine-tuned based on the data that was later used for analysis, it would be helpful to include additional information demonstrating the choices in the optimization procedure are independent of the eventual results. For example, it isn't clear what 'important boundaries being detected late' means, whether that indicates event boundaries were being missed by the original algorithm. Combined with the fact part of the optimization was based on fixing the number of state boundaries to the number of event boundaries – could these choices have increased the chance of finding overlap between state boundaries and event boundaries?

4. Two small notes: the network defined as posterior DMN includes anterior regions, which is slightly confusing; were the regional differences in HRF assessed on the resting state data or the movie watching data?

---

## [Author Response]

[Editors’ note: the authors resubmitted a revised version of the paper for consideration. What follows is the authors’ response to the first round of review.]

The reviewers were in agreement that this is an important topic and that this research is both interesting and promising. However, the reviewers raised a number of significant concerns that centered around two themes. First, there were a number of points raised about the methodology itself and its validity. After some discussion, it was decided that these methodological points could be addressable through additional analysis and/or simulation, but likely require considerably more work than would be usual for an eLife revision. The second set of concerns were with regard to the clear scientific advance over prior work; there was not consensus that these findings move the field forward in a clear way. Reviewer 1 suggested that the generalizability and impact might be improved by drawing direct links to the existing literature, including analysis of a secondary dataset as in the Baldasano et al. (2017) paper. Though, there might be other ways to clarify the impact, as well. Regardless, this is a challenging concern to address in a straightforward way through revision.As addressing these concerns would likely require more than is typically expected for an eLife revision, it was decided to reject this submission. This being said, if you were to undertake the work required to conclusively address these issues, there was sufficient enthusiasm among reviewers that they would be willing to consider this paper again, as a new submission.I have appended the detailed reviews to this decision letter. I hope you find them constructive with this work.Reviewer #1:In this paper, Geerligs et al. focus on the alignment of event boundaries across brain regions. They examine the transitions between brain states using the method introduced by Baldassano et al. (2017), and how these state transitions are shared across nodes of large-scale brain networks. They introduce a method that enables them to map event-timescales in a broader set of regions than previously possible, and they use this method to reveal how functional networks of regions share time-aligned "event transitions".This is a well-written manuscript on a timely and important question.

We thank the reviewer for their compliments about our work.

My main concerns relate to the validity (and potential sources of bias) in the methodology for identifying the event-rate of each region, and I also outline a number of other areas where the conceptual and methodological framing could be improved.p.3 "This dataset, in combination with the application of hyperalignment to optimize functional alignment (Guntupalli et al., 2016), allowed us to study event segmentation across the entire cortex for the first time, because this dataset shows reliable stimulus-driven activity (i.e., significant inter-subject correlations) over nearly all cortical brain regions (Geerligs et al., 2018). "A central methodological question, which affects almost every claim in this manuscript, is whether the inference of event boundaries from the HMM model (the methods in Figure 1) is valid, and in what ways it might be biased. The validity question is simple: does it measure what it is supposed to measure? In particular, I would like the authors to justify the final step, in which they compute the difference between the correlation for real boundaries and the correlation for random boundaries. Surely, this difference computation will be affected by the noise ceiling of the individual ROI being examined? I understand why using the random condition as a "reference" makes some sense, but I do not understand why the final decision is made based on the simple arithmetic difference of the mean value for the random boundaries and real boundaries? I suggest that the authors justify this procedure using a simulation procedure where the ground truth about event transitions is known, and the procedure should be compared against the method applied in the original Baldassano et al. (2017) paper.The bias question is also fairly simple: which factors influence the "k" that is inferred? In particular, if a region has high reliability or low reliability of its response across subjects, does this affect the number of events that will be inferred for that region using the HMM procedure? As noted above, this simulation could additionally investigate how the "k" value varies as function of the noise level (i.e. response reliability) of the ROI.Additionally, although hyperalignment render a larger swathe of cortex available to analysis, but there will still be variability in the reliability of the signal across regions, and this might interact with the hyperalignment performance. In particular, the accuracy of the hyper alignment procedure (for each subject) will presumably also increase for regions whose reliability of response is higher; it is therefore very to consider whether noise (in "space") introduced by the hyperalignment procedure (and varying across regions as a function of their reliability) could further bias the measurement of the event-timescale via the HMM procedure.Finally, to better understand this method, the authors could also apply their approach to the freely available data from the Baldassano et al. (2017) paper. Does this method produce results that are at least qualitatively similar? This could help to resolve the question of why the event timescales in this paper are shorter than those observed in the Baldassano et al. paper.

We thank the reviewer for these valid questions. In trying to answer them, we performed many simulations to determine the validity of our previous approach. These simulations revealed to us that the method we used before was indeed biased by the level of noise in different brain regions. While running these simulations we also discovered some problems with the HMM-approach in dealing with states of unequal length. In the end, all of these issues led us to develop a new method for detecting neural state boundaries and the optimal number of states, which does not suffer from these issues. This new method called greedy state boundary search (GSBS) was used to redo all the analyses in the paper and has now been published in Neuroimage – https://doi.org/10.1016/j.neuroimage.2021.118085.

In the Neuroimage paper, we performed an extensive set of simulations to demonstrate the validity of GSBS. We show that state boundaries can be identified reliably when the data are averaged across groups of ~17/18 or more participants. We also observed that high levels of noise in the data lead our algorithm to over-estimate the number of states in a region. In contrast, regions such as the medial prefrontal cortex, which show relatively low levels of inter-subject synchrony, have a small number of long-lasting neural states. Therefore, we can be confident that the regional differences in the number of states we observe are not due to regional differences in noise.

p.7: Event networks: "We found that event boundaries are shared within long-range networks that resemble the functional networks that are typically identified based on (resting state) timeseries correlations (see figure 3A)".This is one of the most intriguing aspects of this paper. However, it would be much more convincing if the authors would replace their qualitative language (e.g. "resemble") with quantitative metrics of overlap. The overlap could be measure between (a) networks defined based on event-timing and (b) networks defined based on functional connectivity. All of the major functional networks should be available in atlases (e.g. the Yeo lab atlases) or via data sharing repositories. Thus, the authors should be able to substantiate their broad claims of "resemblance" with quantitative demonstrations of how well the event-networks match the functional-connectivity-networks. All of the visual networks as well as the FPN and DMN should be quantitatively compared against standard networks defined elsewhere in the literature.On the same point: p.13 "The fractionation of the DMN into a fast and slow subnetwork closely aligns with the previously observed posterior and anterior DMN subnetworks (Andrews-Hanna et al., 2010; Campbell et al., 2013; Lei et al., 2014)."Again, please quantify the alignment when claiming spatial alignment with prior findings.

To address this concern, we compared the measures of boundary overlap to the functional connectivity observed with correlation of time series across all pairs of searchlights. We show that there is only a medium sized correlation between the two (r=0.39) suggesting that overlap of neural state boundaries is not simply a result of ‘regular’ functional connectivity between brain regions. In fact, we observe that regions that are negatively correlated do tend to have neural state boundaries at the same time.

To determine the correspondence to previously identified networks, we now quantify the spatial alignment between the networks we detected and the networks defined by Power et al. (2011). We also compared the DMN subnetworks to the previously detected DMN subnetwork by Campbell et al. (2013). For the newly identified superior DMN, we were not able to quantify the alignment with the data from Gordon et al. (2020) because those data were in surface space, rather than MNI space.

The new results and comparisons can be found on page 8-12 of the manuscript and tables S1 and S2.

p.13 "Our results show for the first time that neural events are shared across brain regions in distinct functional networks. "The authors should consider re-wording this sentence to distinguish their findings from what was already shown in Figure 4B of Baldassano et al. (2017). In particular, note the commonality of event boundaries across early visual and late visual areas (part of the visual network), as well as the commonality of events across angular gyrus and posterior medial cortex (parts of the DMN).

We have rephrased this to: “In line with previous work (Baldassano et al., 2017) we found that neural state boundaries are shared across brain regions. Our results show for the first time that these boundaries are shared within distinct functional networks.”

On a related note, in the Abstract we read: "This work extends the definition of functional networks to the temporal domain" – I am unclear on how novel this extension is. To the best of my understanding, the concept of dynamic functional connectivity is not new (e.g. Hutchison et al., 2013), and even second-order pattern-transition methods have been employed to study functional networks (e.g. Anzellotti and Coutanche, 2018). I would like the authors to sharpen their argument for why this result is not entirely expected in light of prior work. Shouldn't members of the same functional networks be expected to exhibit state-transitions at rates higher than chance?

We have removed this claim from the abstract. In addition, we now show in the Results section and figure S6 that the correlations between state boundaries cannot be directly explained from the correlations between the average searchlight timeseries (i.e., regular functional connectivity). The following text has been added to the manuscript”:

“Although the networks we identified show overlap with functional networks previously identified in resting state, they clearly diverged for some networks (e.g., the visual network). Some divergence is expected because neural state boundaries are driven by shifts in voxel-activity patterns over time, rather than by the changes in mean activity that we typically use to infer functional connectivity. This divergence was supported by the overall limited similarity with the previously identified networks by Power et al. (2011; adjusted mutual information = 0.39), as well as the differences between the correlation matrix that was computed based on the mean activity time courses in each searchlight and the relative boundary overlap between each pair of searchlights (figure S6; r=0.31). Interestingly, regions with strongly negatively correlated mean activity time courses typically showed overlap that was similar to or larger than the overlap expected by chance. Indeed, the relative boundary overlap between each pair of searchlights was more similar to the absolute Pearson correlation coefficient between searchlights (r=0.39) than when the sign of the correlation coefficient was preserved (r=0.31). This suggests that pairs of regions that show negatively correlated BOLD activity still tend to show neural state boundaries at the same time.”

p.11. I struggled to follow the logic of the analysis employed in Figure 6. Why is event duration being predicted from individual frequency bands of the PSD? There is voluminous evidence for band-specific and region-specific artifact (e.g. Birn et al., 2013; Shmueli et al., 2007). Furthermore, distinct functional networks have distinct frequency profiles and coherence patterns (e.g. Salvador et al., 2008; Baria et al., 2011; Stephens et al., 2013). Finally, the frequency bands in the PSD are non-independent (because of the temporal smoothing in the BOLD signal). Therefore, the relationship between frequency band and event duration is confounded by (i) non-independence of frequencies and (ii) frequency covariation across brain regions which arises for a multitude of reasons. The results in Figure 6A seem rather noisy to me, and I imagine that this is because the regression procedure on the PSD is influenced by many interacting and confounding variables.Another region why this analysis produces (in my opinion) curious results is that it spans distinct sensory modalities which are already known to have opposite PSD-event relationships: along the auditory pathway, PSDs get flatter as event time-scales get longer, while in the visual pathway, PSDs in V1 are already very steep, even while the event timescales are short. It is not clear what is gained by fitting a single model to regions with obviously different relationships of PSD and event structure.

In response to the comments of reviewers 1 and 3, we have removed the resting state analyses from the paper. We realised that using the power spectral density as a proxy for neural state timescales is suboptimal, given the variable duration of neural states within brain regions. In response to reviewer suggestions, we have shifted the focus of the manuscript to what neural states can tell us about the neural mechanisms underlying event segmentation.

p.12. "These results suggest that visual and auditory stimulation are a prerequisite for observing the temporal hierarchy we describe in this paper and that this hierarchy only partly reflects an intrinsic property of brain function that is also present in the resting state."I do not follow the logic supporting this claim. How can we know whether the (event-based) temporal hierarchy is preserved in the resting state unless we can measure the event transitions in the resting state data? Isn't this analysis just another way of saying that the PSDs have different shapes during rest and during movie viewing?

This claim has been removed from the paper, in relation to the previous point made by the reviewer.

Reviewer #2:In this paper, Geerlings and colleagues leverage a large, publicly-available dataset in order to assess shared and distinct timescales of neural pattern shifts at event boundaries across different areas of the brain. In line with prior work, the authors report a gradient of timescales in neural event segmentation, with sensory regions comprising the fastest-shifting areas and 'default mode' nodes such as precuneus and medial prefrontal cortex comprising the slowest-shifiting areas. Importantly, the authors build on this previous research and demonstrate that canonical functional networks – such as the frontoparietal network, and the 'default mode' network – feature distinct subnetworks with corresponding faster and slower timescales of pattern shifts. Finally, a fairly novel analysis applied to these types of data examined power spectral density across regions, which could be used to predict event duration across regions (consistent with observed pattern shifts), and could partly, but not entirely, characterize resting-state fMRI data (suggesting that the audiovisual stimulus drove additional functional properties in brain networks not observed during rest).Overall, this is an interesting and timely study. The question of how the brain segments naturalistic events is one of increasing popularity, and this manuscript approaches the question with a large sample size and fairly thorough analyses. That said, there are a number of questions and concerns, primarily regarding the analyses.

We thank the reviewer for their positive feedback.

• Procedures such as hyperalignment, or the related shared response model used by Baldassano and colleagues, are typically implemented by training on one set of the data, and applying the alignment procedure to a separate, held-out dataset (i.e., training and testing sets). It is unclear whether this approach was taken in the current study, or whether the hyperalignment algorithm was trained and tested on same dataset. In the latter case, there is a degree of circularity in the way across-participant alignment was conducted, potentially leading to biased correlation measures. The movie used in the CamCAN dataset is only 8 minutes long, which is probably not enough data for obtaining separate training and test datasets. However, this is still potentially a serious issue for this manuscript, and I am not sure if the use of hyperalignment is appropriate. If I have misunderstood the methodology, it perhaps warrants some clarification in how the training and application of the hyperalignment algorithm proceeded. (I will note that I am aware you used cross-validation for deriving the number of events, but that is unfortunately a separate issue from a train-test split in the hyperalignment routine itself.)

We agree with the reviewer that hyperalignment parameters are typically estimated in a separate dataset. Hyperalignment can introduce dependencies between datasets from different participants, which can result in biased statistics. To avoid this issue, we ran hyperalignment separately in each of the participant subgroups reported in the manuscript. All statistical testing was performed on independent subgroups of participants (17/18 participants per group), which were hyperaligned separately (i.e. within each group).

• A key finding from the study is that the FPN and DMN fractionate into different subnetworks that have fast and slow timescales. As noted above, the present results are based on an analysis of data from a relatively short period of time. Although the sample size is very large, one wonders whether this distinction would remain solid with a longer movie. With a very short movie, one can only sample a small number of real events, and this could lead to some instability in estimates of the timescale of representations in relation to the events. This might be an issue in relation to the differentiation of fast and slow subnetworks within the FPN and DMN. For instance, Figure 3B, suggests that the fit values for the slow FPN remain more or less stable across a range of event durations (which presumably reflect k values?). The slow FPN shows an interesting bimodal distribution (as do many of the networks) with the second peak coinciding with the peak for the fast FPN. The differentiation is a bit more convincing for the fast and slow DMN, but it is still not clear whether there are enough events and enough fMRI data from each subject to ensure reliable estimates of the timescales. Just to provide some context for this point, some estimates suggest that reliable identification of resting state networks requires at least 20 minutes of fMRI data.

To illustrate the reliability of our approach in identifying regional timescale differences, we now estimate the timescales separately in two independent samples of participants (see figure 1). The correlations between the results of these two groups is very high at the voxel level (r=0.85), suggesting that even in this short dataset we can reliably estimate regional differences in timescale.

Second, figure 4 shows that the timescale differences across the different DMN subnetworks are highly reliable across the searchlights in these networks. It should be noted that due to substantial improvements to our data analysis pipeline, we no longer observe two distinct FPN networks.

• Throughout the paper, fMRI results are described in reference to event processing, but the relationship is underdeveloped. Much of the paper relies on the Hidden Markov Model, which assumes that there is a pattern that remains stationary throughout an event. Baldassano's data shows a surprisingly strong correspondence in posterior medial cortex, but it is less clear whether this assumption is valid for other areas. In relation to this point, one can think of event processing as an accumulation of evidence. At the onset of an event, one might have a decent idea of what is about to happen, but as information comes in, the event model can be refined to make stronger predictions. These kinds of within-event dynamics would be lost in the Hidden Markov model. A related point is that the paper conflates timescales of neural states with psychologically meaningful conceptions of events. EST suggests that event segmentation is driven by prediction error-by one interpretation of the model, sensory information can change considerably without leading one to infer an event boundary. However, change in incoming sensory information would almost certainly lead to the detection of "event boundaries" across short timescales in sensory cortical areas. Figure 5 makes it fairly clear that there is a pretty strong distinction to be made between data-driven event identification based on the fMRI data and psychologically meaningful events inferred by the subjects. It would be helpful for the authors to be more clear about what the data do and do not show in relation to putative event cognition processes.

We strongly agree with the reviewer that it is important to distinguish between the transitions in brain activity patterns observed in the current paper and perceived event boundaries that have been described extensively in the behavioural literature. We have therefore changed the terminology in the paper and refer to ‘neural states’, rather than events when referring to the brain data. We now also discuss much more extensively what our findings can tell us about the mechanisms underlying event segmentation in both the introduction and Discussion sections.

• Why were voxels with an intersubject correlation of less than r=0.35 excluded from analyses? Is this based on prior studies or preliminary analyses? It is not necessarily a bad thing if this choice was made arbitrarily, but I imagine this threshold could have important impacts on the data as presented, so it is worth clarifying.

We investigated the effect of thresholding based on inter-subject correlation in a recent Neuroimage paper (Geerligs et al., 2021) and we observed that it did not result in more reliable estimates of neural state boundaries. Hence, we have removed this threshold from the analysis pipeline.

• Was ME-ICA the only step taken to account for head motion artifacts? If so, there is some concern about whether this step was sufficient to deal with the potential confound. This is especially critical given the fairly brief time series being analyzed here. It would be more compelling to see a quantitative demonstration that head motion is not correlated with the measures of interest.

ME-ICA denoising is currently the most effective method to deal with head motion (Power et al., 2018, PNAS). In addition, all our analyses are performed on group averaged data. This means that head motion that is not synchronized across participants cannot affect the results. To further investigate this potential confound, we computed the correlation between scan to scan head motion and state boundaries for each searchlight within each of the 15 groups of participants. Next we used a Wilcoxon signrank test to investigate if these correlations were significantly different from zero across the 15 groups. We found that none of the searchlights showed a significant association between the average head motion in each group of participants and the occurrence of neural state boundaries. These results are reported in the supplementary Results section. Together these results suggest that head motion did not confound the results reported in the current manuscript.

• A related issue is that of eye movements. Eye movements are related to event processing (e.g., Eisenberg et a., 2018), so one can expect neural activity related to event prediction/prediction error to be confounded with lower-level effects related to eye movements. For instance, we might expect signal artifacts in the EPI data, as well as neural activity related to the generation of eye movements, and changes in visual cortex activity resulting from eye movements. It is unlikely that this issue can be conclusively addressed with the current dataset, and it's not a deal-breaker in the sense that eye movements are intrinsically related to naturalistic event processing. However, it would be useful for the authors to discuss whether this issue is a potential limitation.

We agree with the reviewer that eye movements may affect neural data in the frontal eye fields as well as sensory cortices. However, they are indeed intrinsically related to naturalistic stimulus processing. Fixating the eyes would result in a very unnatural mode of information processing which might bias neural activity in very different ways. In response to this comment from the reviewer, we added the following section to the discussion:

“It should be noted that this more naturalistic way of investigating brain activity comes at a cost of reduced experimental control (Willems et al., 2020). For example, some of the differences in brain activity that we observe over time may be associated with eye movements. Preparation of eye movements may cause activity changes in the frontal-eye-fields (Vernet et al., 2014), while execution of eye movements may alter the input in early sensory regions (Lu et al., 2016; Son et al., 2020). However, in a related study (Davis et al., 2021), we found no age difference in eye movement synchrony while viewing the same movie, despite our previous observation of reduced synchrony with age in several areas (particularly the hippocampus, medial PFC, and FPCN; Geerligs et al., 2018), suggesting a disconnect between eye movements and neural activity in higher-order areas. In addition, reducing this potential confound by asking participants to fixate leads to an unnatural mode of information processing which could arguably bias the results in different ways by requiring participants to perform a double task (monitoring eye movements in addition to watching the movie). ”

• The power spectral analyses were a bit difficult to follow, but more importantly, the motivation for the analysis was not clearly described. The main take home points from this analyses are nicely summarized at the end of p. 14, but it would be helpful to clarify the motivation for this analysis (and the need for doing it) on p.11 in the Results section. Relatedly, is Figure 6A an example spectrum from a particular voxel or region, or an average across regions?

These analyses have now been removed from the paper. Based on the comments from reviewers 1 and 3, we realised that using the PSD as a proxy for neural state timescales is suboptimal, given the variable duration of neural states within brain regions.

• The take-home message appears to be that different brain networks have different timescales at which they seem to maintain event representations. Moreover, certain networks (e.g., the posterior medial/'default mode' network) do not have uniformly fast or slow timescales. The network-based analysis used here is indeed novel, but the impact of the work could be enhanced by clarifying the significance of the results in relation to what we know about event processing. The explicit demarcation of 'fast' and 'slow' subnetworks may be the key conceptual advance, as was the power spectral analysis, but it isn't clear whether these conclusions could also be ascertained from the maps shown in Baldassano et al., 2017 or other papers from the Hasson group.

We have completely rewritten the introduction and Discussion sections to clarify the significance of our work in relation to what we know about event processing.

This review was completed by Zach Reagh, Ph.D. in collaboration with Charan Ranganath, Ph.D. (I sign all reviews)Reviewer #3:Geerligs and colleagues conduct a thorough set of analyses aimed at identifying event segmentation timescales across the cortex in a large cohort of participants. They extend previous work by Baldassano et al. by covering the entire cortex, and nicely control for the power spectrum of different regions. In addition, they examine which regions share the same event boundaries, not just the same timescale, and relate these to functional connectivity networks. Overall, their work is impressive and rigorous, but there are a few points that make it somewhat difficult to assess the how strong the contribution is to our understanding of processing timescales:

Thank you for your enthusiasm. We address your specific concerns below.

1. The authors divide the brain into functional networks based on boundary similarity and find that this division is very similar to functional networks defined using resting-state timeseries correlations. They further find increased similarity between regions of different networks that are that are interconnected. Wouldn't the similarity between boundary vectors be strongly linked to the timeseries correlations (both between regions in the same network and across networks)? While the similarity-based functional networks aren't completely identical to those identified in rest, perhaps the same results would be obtained by correlating timeseries in this specific dataset, using the movie data (altering the interpretation of the results).

We now show in the Results section and figure S6 that the correlations between state boundaries cannot be directly explained from the correlations between the average searchlight timeseries (i.e., regular functional connectivity). Although there are similarities between the two, as we would expect, the correlation between them is only r=0.31, suggesting that the same results could not have been obtained using ‘regular’ functional connectivity analyses in the movie dataset. The following text has been added to the manuscript:

“Although the networks we identified show overlap with functional networks previously identified in resting state, they clearly diverged for some networks (e.g., the visual network). Some divergence is expected because neural state boundaries are driven by shifts in voxel-activity patterns over time, rather than by the changes in mean activity that we typically use to infer functional connectivity. This divergence was supported by the overall limited similarity with the previously identified networks by Power et al. (2011; adjusted mutual information = 0.39), as well as the differences between the correlation matrix that was computed based on the mean activity time courses in each searchlight and the relative boundary overlap between each pair of searchlights (figure S6; r=0.31).

Interestingly, regions with strongly negatively correlated mean activity time courses typically showed overlap that was similar to or larger than the overlap expected by chance. Indeed, the relative boundary overlap between each pair of searchlights was more similar to the absolute Pearson correlation coefficient between searchlights (r=0.39) than when the sign of the correlation coefficient was preserved (r=0.31). This suggests that pairs of regions that show negatively correlated BOLD activity still tend to show neural state boundaries at the same time.”

2. It seems that the power spectrum analysis is run both on the resting-state data and on the movie data, whereas the timescale segmentation is run only on the movie data. I expect this is because hyperalignment is possible only when using a shared stimulus, and the HMM is run only on the hyperaligned data. However, this may bias the correlations presented in figure 6 – the movie PSD-based timescale estimation would be expected to be more similar to the HMM timescales than the rest, simply because the same data is used. A more convincing analysis would be to run the HMM on the rest data as well, and test for correlations between the two estimations of event timescales in the rest data, although this would entail substantial additional analyses (as HMM would also have to be run on non-hyperaligned movie data for comparability). It would also help with point 1, testing whether similarity in boundary vectors arises directly from timeseries correlations. I realize this adds quite a bit of analysis, and the authors may prefer to avoid doing so, but the conclusions arising from the power spectrum analysis should be softened in the Results and Discussion, clearly mentioning this caveat.

Unfortunately, it is not possible to detect neural states in the resting state data, since detecting neural states requires data averaging across participants. Because resting-state fluctuations in brain activity are not stimulus driven, data cannot be meaningfully averaged across participants in resting state. Therefore, based in the comments from reviewers 1 and 3 and the altered focus of the paper on neural mechanisms underlying event segmentation, we have removed the analyses from the PSDbased timescale estimation from the paper.

3. It would aid clarity to better separate the current contributions from previous findings, in the Results, and mainly in the Discussion. The authors do describe what has previously been found, citing all relevant literature, but it would be helpful to have a clear division of previous findings and novel ones. For example in the first paragraph of the Discussion, and in general when discussing the interpretation of activity the different regions (currently regions that have already been found are somewhat intermixed with the new regions found).

We have rewritten the introduction and Discussion sections extensively to make more clear what is novel in the current study.

[Editors’ note: what follows is the authors’ response to the second round of review.]

The manuscript has been improved but there are some remaining issues that need to be addressed, as outlined below:The reviewers were positive about the revisions you made to this submission and felt that extensive work had been done to improve the paper. There were a few remaining points raised by this review that could be addressed the further strengthen the paper. The Reviewing Editor has drafted this to help you prepare a revised submission.

We are very happy to hear that the reviewers were positive about the extensive work we did for the revision. We have taken care to address all remaining points. A summary of the most important changes is provided below.

First, we have described our analyses more clearly, regarding how the overlap between neural state and event boundaries is computed and how we compare this overlap for shared vs. non-shared states. This also helped us streamline our analyses more and we now use the same metric (absolute overlap) throughout the manuscript, also for investigating the effect of boundary strength. This should make the paper easier to follow for readers. Second, we have added additional analyses to more clearly demonstrate the effects of boundary sharing across (large) parts of the cortical hierarchy in relation to perceiving event boundaries. These analyses provide stronger support for the claims we made in the previous version of the manuscript. Finally, we have added some analyses to make sure that the effects we see cannot be explained by shared confounds or noise.

Essential revisions:1. Reviewer 1 has raised some additional points for clarification in their review, as noted below. These should be clarified in a revision. Please refer to the comments below for these notes.

We have clarified all the points that the reviewer raised. More details about these clarifications are provided in the answers to specific reviewer comments below.

2. Some of the conclusions do not completely reflect the results. If additional analyses are not added, perhaps these conclusions could be rephrased, such as "some of the neural boundaries are represented throughout the hierarchy.… until eventually reflected in conscious experience" (p. 14) and "boundaries that were represented in more brain regions at the same time were also more likely to be associated with the experience of an event boundary" (p. 15).

We rephrased the first sentence (originally p 14) to: “This finding suggests that some of the neural state boundaries that can be identified in early sensory regions are also consciously experienced as an event boundary. Potentially because these boundaries are propagated to regions further up in the cortical hierarchy.”

The second sentence (p. 15) is now supported more strongly by the results that we have obtained through new analyses. More details about these new results are provided in the answers to specific reviewer comments below.

3. Since the GSBS algorithm was fine-tuned based on the data that was later used for analysis, it would be helpful to include additional information demonstrating the choices in the optimization procedure are independent of the eventual results. For example, it isn't clear what 'important boundaries being detected late' means, whether that indicates event boundaries were being missed by the original algorithm. Combined with the fact part of the optimization was based on fixing the number of state boundaries to the number of event boundaries – could these choices have increased the chance of finding overlap between state boundaries and event boundaries?

The primary concern with the performance of the original algorithm in some brain regions, was that the placement of one new boundary resulted in a huge increase in the t-distance. This suggests there was a strong neural state boundary (i.e. large transition in brain activity patterns) that was detected in a late iteration of the algorithm. Therefore, the peak of the t-distance was at a high value of *k* (number of states) and the inferred optimal number of states was higher than it should have been. By placing two boundaries at the same time (i.e. inferring the location of a state, rather than the location of one boundary), the algorithm behaves much more stable and no longer shows this kind of behavior.

We have not investigated whether those boundaries that are detected late tend to overlap with event boundaries. This is because it is not the problem that the original algorithm missed boundaries, but rather that too many boundaries were added, probably including many boundaries that did not overlap with events. We have now clarified this point in the methods section:

“First, GSBS previously placed one boundary in each iteration. We found that for some brain regions, this version of the algorithm showed sub-optimal performance. A boundary corresponding to a strong state transition was placed in a relatively late iteration of the GSBS algorithm. This led to a steep increase in the t-distance in this particular iteration, resulting in a solution with more neural state boundaries than might be necessary or optimal (for more details, see the supplementary methods and figure 1A in appendix 1).”

And in the supplementary methods in appendix 1:

“First, we discovered that for specific brain regions the original GSBS algorithm performed suboptimally; the placement of one new boundary at a late stage in the fitting process resulted in a large increase in the t-distances (our measure of fit; see appendix 1 – figure 1A). This suggests that a strong neural state boundary (i.e. demarcating a large change in neural activity patterns) was detected only in a late iteration of the algorithm, which led to an overestimation of the number of neural states.”

For completeness we also reran the analyses comparing states and events without fixing the number of states to the number of events. The results of these analyses support our original conclusions and are shown in appendix 1 – figure 2. The adapted text in the supplementary methods section of appendix 1 is copied below:

“To investigate how these changes to GSBS impacted reliability, we split the data in two independent groups of participants and looked at the percentage of overlapping boundaries between the groups for each searchlight. To make sure differences in number of states between methods did not impact our results, we fixed the number of state boundaries to 18 or 19. Because the states-GSBS algorithm can place one or two boundaries at a time, we cannot fix the number of state boundaries exactly, which is why is can be either 18 or 19. We found that the number of overlapping boundaries between groups was substantially higher for states-GSBS, compared to the original GSBS implementation and also compared to the GSBS implementation with altered finetuning (see appendix 1 – figure 2A). This was also the case when we used the optimal number of states as determined by the t-distance, instead of fixing the number of states (see appendix 1 – figure 2B).”

4. Two small notes: the network defined as posterior DMN includes anterior regions, which is slightly confusing; were the regional differences in HRF assessed on the resting state data or the movie watching data?

We have added the following sentence to clarify the issue about network naming: “It should be noted that all three of the DMN subnetworks include some anterior, superior and posterior subregions; the names of these subnetworks indicate which aspects of the networks are most strongly represented.”

Regional differences in HRF were assessed with movie data. This has now been clarified in the methods section:

“We also investigated the effects of estimating the HRF shape based on the movie fMRI data, instead of using the canonical HRF and found that this did not have a marked impact on the results (see supplementary methods in appendix 1).”

And also in the supplementary methods in appendix 1: “To investigate whether such differences might impact our results, we estimated the HRF for each participant and each searchlight, using the rsHRF toolbox that is designed to estimate HRFs in resting state data (Wu et al., 2021). In this case we applied the algorithm to our fMRI data recorded during movie watching.”

Additional Suggestions for Revision (for the authors):One of the reviewers had some suggestions for additional analyses that might strengthen the results. We pass them along to you here, but you should view these as optional. Only include them if you agree that they will strengthen the conclusions.There are a few analyses that may help strengthen the conclusions - these are suggested as optional additional analyses, but the authors should feel free not to include them:• To verify the overlap between searchlights is not due to various artifacts, it may be preferable to compare the searchlight in one region with the searchlights of other groups in the second region (following the rationale of intersubject functional connectivity vs. functional connectivity).

To address this point, we have repeated the overlap analyses in data with two independent groups of participants, like the reviewer suggested. The results of these analyses are described in the results section and in appendix 1 - figure 6.

“To make sure the observed relative boundary overlap between searchlights was not caused by noise shared across brain regions, we also computed the relative boundary overlap across two independent groups of participants (similar to the rationale of inter-subject functional connectivity analyses Simony et al., 2016). We observed that the relative boundary overlap computed in this way was similar to the relative overlap computed within a participant group (r=0.69; see appendix 1 - figure 6), suggesting that shared noise is not the cause of the observed regional overlap in neural state boundaries.”

And in the supplementary results section of appendix 1:

• It would also be interesting to further explore the nature of the overlap - to see whether there are specific state boundaries that drive most of the overlap or whether different pairs of regions have different overlapping boundaries. This could be used to explore the nature of the hierarchy between regions, beyond just finding that higher regions share boundaries with lower regions. For example, it could enable testing whether state shifts shared by multiple lower level regions are the ones that traverse the hierarchy.

To further explore the nature of the overlap we have now added an additional analysis in which we clustered time points with similar patterns of neural states. Below we copied the description from the results section:

“So far, we have focused on comparing state boundary timeseries across regions or between brain regions and events. However, that approach does not allow us to fully understand the different ways in which boundaries can be shared across parts of the cortical hierarchy at specific points in time. To investigate this, we can group timepoints together based on the similarity of their boundary profiles;

i.e. which searchlights do or do not have a neural state boundary at the same timepoint. We used a weighted stochastic block model (WSBM) to identify groups of timepoints, which we will refer to as ‘communities’. We found an optimal number of four communities (see figure 7). These communities group together timepoints that vary in the degree to their neural state boundaries are shared across the cortical hierarchy: timepoints in the first community show the most widely spread neural state boundaries across the hierarchy, while timepoints in the later communities show less widespread state transitions. We found that from community 1 to 4, the prevalence of state boundaries decreased for all networks, but most strongly for the FPCN and CON, sDMN, aDMN and auditory networks. However, the same effect was also seen in the higher visual and SMN and motor networks. This might suggest that boundaries that are observed widely across lower-level networks are more likely to traverse the cortical hierarchy. We also found a similar drop in prevalence of event boundaries across communities, supporting our previous observation that the perception of event boundaries is associated with the sharing of neural state boundaries across large parts of the cortical hierarchy. We repeated this analysis in two independent groups of participants to be able to assess the stability of this pattern of results. Although group 1 showed an optimum of four communities and group 2 an optimum of five communities, the pattern of results was highly similar across both groups (see appendix 1 - figure 8).”

• Further to this, it would be interesting to test whether event boundaries and non-event neural state boundaries form a similar hierarchy (though this may not be feasible with such a low number of event boundaries).

Unfortunately, this was indeed not feasible given the low number of event boundaries in our data. A longer movie would be required to answer this question.

• To assess the effects of noise reduction on the overlap between neural state boundaries and event boundaries, it may be worth testing whether neural state boundaries shared across groups of participants are also more likely to be event boundaries (and specifically whether this effect is stronger in the same regions arising from the co-occurrence analysis). This analysis wouldn't provide an answer, but could help shed some light on the role of noise reduction.

We have performed additional analyses to test whether these effects could indeed be due to noise reduction. These analyses have shown that noise reduction is an unlikely cause of our results. The relevant parts of the results section and supplementary results in appendix 1 are copied below:

Results section:

“Analyses shown in the supplementary results section in appendix 1 demonstrate that these increases in overlap for shared vs. non-shared boundaries cannot be attributed to effects of noise (see also appendix 1 - figure 7).”

Supplementary results in appendix 1:

“Effects of noise on overlap between neural states and events for shared boundaries

One concern is that identifying boundaries shared by two regions has a similar effect to averaging, which provides a better estimation of boundaries within each searchlight because it reduces noise. This noise reduction could be the cause of the increased overlap between events and neural states for shared boundaries vs. non-shared boundaries. To investigate this possibility we examined the increase in overlap for shared vs. non-shared values in the data averaged across 265 participants as well as for each independent subgroup of 17/18 participants. If noise reduction is the cause of the increase in overlap with event boundaries, we should expect the difference between shared and nonshared boundaries to be largest in the smaller independent subgroups where there is the most to be gained from noise reduction. In contrast, if the increase in overlap with event boundaries is a real effect, not due to noise, its effect size should be larger in the data averaged across all participants, where estimates of boundary locations are more accurate. The results in appendix 1 - figure 7 show that the latter interpretation is correct, making it unlikely that the observed increase in overlap between neural state and event boundaries is related to noise.”

Reviewer #1:This work investigates timescales of neural pattern states (periods of time with a relatively stable activity pattern in a region) across the brain and identify links between state shifts and perceived boundaries events. In multiple regions, they find significant overlap between state shifts and event boundaries, and an even stronger overlap for state shifts that occur simultaneously in more than one region. The results are interesting and timely and extend previous work by Baldassano et al. that found a similar hierarchy in a specific set of brain regions (here extended to the entire cortex).StrengthsThe question of whether neural state shifts form a hierarchy such that state shifts in higher regions coincide with state shifts in sensory regions, and the question of whether event boundaries occur at conjunctions of shifts in different regions are both very interesting.The optimized GSBS method nicely overcomes limitations of previous methods, as well as a previous version of GSBS. In general, justification is provided for the different analysis choices in the manuscript.The current work goes beyond previous work by extending the analysis to the entire cortex, revealing that state shifts in higher regions of the cortex overlap with state shifts in lower regions of the hierarchy.WeaknessesOne of the important conclusions of the paper is that simultaneous neural state shifts in multiple brain regions are more likely to be experienced as boundaries. This finding fits in nicely with existing literature, but the analysis supporting it is not as compelling as the rest of the analyses in the paper:1. The methods section describing the analysis is not entirely clear. Do Oi, Oj refer to the number of neural state boundaries in searchlights I,j? Or the number of neural state boundaries in each that overlap with an event boundary? If the former (which was my initial interpretation), then how is the reference searchlight chosen – max {Oi,Oj}, as indicated by the formula, or the searchlight with the larger overlap of its unique boundaries (and is the overlap calculated in numerical value or the proportion of overlap)? Given the unclarity, it is difficult to assess whether the degree of overlap between neural state boundaries and event boundaries in each of the searchlights (and/or the number of boundaries in each) could affect the results. It would be helpful to provide verification (either mathematically or with simulations) that higher overlap in one/both searchlights does not lead to a larger difference in overlap between shared and non-shared boundaries.

We agree that our initial description of this analysis was not sufficiently clear. We have now clarified the description of the approach we used. In the previous version of the paper, we used the relative boundary overlap to quantify the overlap for both the shared and non-shared boundaries, but after some simulations we did based on your suggestions, we realized that this metric was biased against pairs of regions with a high number of states (higher than the number of events). That is why we now use the absolute overlap in our analysis, which only depends on the proportion of neural state boundaries that overlap with events. This also led to much stronger evidence for the increased overlap between shared vs. non-shared neural state boundaries with event boundaries.

We have extensively revised our mathematical descriptions of both the overlap metrics as well as our explanations of how we computed the overlap difference between shared/non-shared pairs. These new descriptions clarify that higher overlap in one/both searchlights does not lead to a larger difference in overlap between shared and non-shared boundaries. The overlap metric only depends on the proportion of neural states that overlap with an event boundary. If that proportion is the same for shared/non-shared boundaries then the absolute overlap will also not show any difference.

The relevant sections of text are copied below:

“Comparison of neural state boundaries to event boundaries

To compare the neural state boundaries across regions to the event boundaries, we computed two overlap metrics; the absolute and relative boundary overlap. Both overlap measures were scaled with respect to the expected number of overlapping boundaries. To compute these values, we define E as the event boundary timeseries and S_i_ as the neural state boundary timeseries for searchlight i. These timeseries contain zeros at each timepoint t when there is no change in state/event and ones at each timepoint when there is a transition to a different state/event.

The overlap between event boundaries and state boundaries in searchlight i is defined as:Oi=∑t=1nEt,∙Si,t

where n is the number of TRs.

If we assume that there is no association between the occurrence of event boundaries and state boundaries, the expected number of overlapping boundaries is defined as in Zacks et al. (2001a) as:OEi=1/n∙∑t=1nEt∙∑t=1nSi,t

Because the number of overlapping boundaries will increase as the number of state boundaries increases, the absolute overlap (OA) was scaled such that it was zero when it was equal to the expected overlap and one when all neural state boundaries overlapped with an event boundary. The absolute overlap therefore quantifies the proportion of the neural state boundaries that overlap with an event boundary:OAi=Oi−OEi∑t=1nSi,t−OEi.

Instead, the relative overlap (OR) was scaled such that is was one when all event boundaries overlapped with a neural state (or when all neural state boundaries overlapped with an event boundary if there were fewer state boundaries than event boundaries). In this way, this metric quantifies the overlap without penalizing regions that have more or fewer state boundaries than event boundaries. The relative overlap is defined as:ORi=Oi−OEimin{∑t=1nEt,∑t=1nSi.t}−OEi

And later in the methods section:

“To look in more detail at how boundaries that are shared vs. boundaries that are not shared are associated with the occurrence of an event boundary, we performed an additional analysis at the level pairs of searchlights. For each pair of searchlights i and j, we created three sets of neural state boundaries timeseries, boundaries unique to searchlights i or j: Si,∼j and Sj,∼i and boundaries shared between searchlights i and j: Si&j. More formally, using the binary definition of the neural state boundary timeseries *S*_*i*_ and *S*_*j*_, these are defined at each timepoint t as:Siandj,t=Si,t∙Sj,t,Si,∼j,t=Si,t−Siandj,t,Sj,∼i,t=Sj,t−Siandj,t.

Then, we investigated the absolute overlap between each of these three boundary series and the event boundaries as described in the section ‘Comparison of neural state boundaries to event boundaries’. This resulted in three estimates of absolute boundary overlap; for boundaries unique to searchlight *i* (OAi,∼j) and searchlight *j* (OAj,∼i) and the shared boundaries (OAi&j). Then we tested whether the absolute overlap for the shared boundaries was larger than the absolute overlap for non-shared boundaries, using the searchlight that showed the largest overlap in their unique boundaries as the baseline: OAi&j>max{OAi,∼j,OAj,∼i}. Because the absolute boundary overlap is scaled by the total number of neural state boundaries, it is not biased when there is a larger or smaller number of shared/non-shared states between searchlights *i* and *j*. It is only affected by the proportion of neural state boundaries that overlap with an event boundary. If that proportion is the same for shared and non-shared boundaries, the overlap is also the same.”

In the Results section:

“To investigate the role of boundary co-occurrence across networks in more detail, we investigated for each pair of searchlights whether boundaries that are shared have a stronger association with perceived event boundaries as compared to boundaries that are unique to one of the two searchlights. We found that boundary sharing had a positive impact on overlap with perceived boundaries, particularly for pairs of searchlights within the auditory network and between the auditory network and the anterior DMN (see figure 6A and B). In addition, we saw that neural state boundaries that were shared between the auditory network and the early and late visual networks, and the superior and posterior DMN were more likely to be associated with a perceived event boundary than non-shared boundaries. The same was true for boundaries shared between the anterior DMN and the lateral and medial SMN network and the posterior DMN. Boundary sharing between the other higher-level networks (pDMN, sDMN, FPCN and CON) as well as between these higher-level networks and the SMN networks was also beneficial for overlap with event boundaries. On a regional level, the strongest effects of boundary sharing was observed in the medial prefrontal cortex, medial occipital cortex, preceuneus, middle and superior temporal gyrus and insula (see figure 6B). Analyses shown in the supplementary Results section in appendix 1 demonstrate that these increases in overlap for shared vs. non-shared boundaries cannot be attributed to effects of noise (see also appendix 1 – figure 7).”

2. The analysis focuses on pairs of searchlights/regions, demonstrating that in a subset of regions there is a higher chance of an overlap with event boundaries for neural state boundaries that are shared between two regions. Yet the interpretation goes beyond this, suggesting that "boundaries that were represented in more brain regions at the same time were also more likely to be associated with the experience of an event boundary". Additional analyses would be needed to back this claim, demonstrating that overlap between a larger number of regions increases the chance of perceiving a boundary.

We have performed two additional analyses that provide strong support for this conclusion. First, we modified the analyses to investigate co-occurrence within networks and across the whole brain in figure 5A. Second, we added an exploratory analysis in which we identify communities of time points that also supports this claim (see figure 7). The relevant sections from the Results section are copied below:

“Shared neural state boundaries and event boundaries

Previous research on event segmentation has shown that the perception of an event boundary is more likely when multiple features of a stimulus change at the same time (Clewett et al., 2019). When multiple sensory features changes at the same time, this could be reflected in many regions within the same functional network showing a state boundary at the same time (e.g. in the visual network when many aspects of the visual environment change), or in neural state boundaries that are shared across functional networks (e.g. across the auditory and visual networks when a visual and auditory change coincide). Similarly, boundaries shared between many brain regions within or across higher-level cortical networks might reflect a more pronounced change in conceptual features of the narrative (e.g. the goals or emotional state of the character). Therefore, we expect that in a nested cortical hierarchy, neural state boundaries that are shared between many brain regions within functional networks, and particularly those shared widely across functional networks, would be more likely to be associated with the perception of an event boundary. To investigate this, we first weighted each neural state boundary in each searchlight by the proportion of searchlights within the same network that also showed a boundary at the same time. This is very similar to how we investigated the role of boundary strength above.”

Recommendations for the authors:• It would be interesting to test whether event boundaries and non-event neural state boundaries form a similar hierarchy (though this may not be feasible with such a low number of event boundaries).

Unfortunately, this was indeed not feasible given the low number of event boundaries in our data. A longer movie would be required to answer this question.